# Immune genes are hotspots of shared positive selection across birds and mammals

**Allison J Shultz[1,2,3]†\*, Timothy B Sackton[1]\***

[1]Informatics Group, Harvard University, Cambridge, United States; [2]Department of Organismic and Evolutionary Biology, Harvard University, Cambridge, United States; [3]Museum of Comparative Zoology, Harvard University, Cambridge, United States

**Abstract** Consistent patterns of positive selection in functionally similar genes can suggest a common selective pressure across a group of species. We use alignments of orthologous protein-coding genes from 39 species of birds to estimate parameters related to positive selection for 11,000 genes conserved across birds. We show that functional pathways related to the immune system, recombination, lipid metabolism, and phototransduction are enriched for positively selected genes. By comparing our results with mammalian data, we find a significant enrichment for positively selected genes shared between taxa, and that these shared selected genes are enriched for viral immune pathways. Using pathogen-challenge transcriptome data, we show that genes up-regulated in response to pathogens are also enriched for positively selected genes. Together, our results suggest that pathogens, particularly viruses, consistently target the same genes across divergent clades, and that these genes are hotspots of host-pathogen conflict over deep evolutionary time.

DOI: https://doi.org/10.7554/eLife.41815.001

**\*For correspondence:**
ashultz@nhm.org (AJS);
tsackton@g.harvard.edu (TBS)

**Present address:** †Natural History Museum of Los Angeles County, Los Angeles, United States

**Competing interests:** The authors declare that no competing interests exist.

## Introduction

Central to the study of evolutionary biology is the desire to understand how natural selection operates across a diverse set of populations and species. While many selective pressures vary across taxa, some common selective pressures may result in consistent patterns of natural selection across a set of species. By taking an unbiased approach and scanning as many identifiable orthologous genes as possible across a set of species for signatures of positive selection, it may be possible to identify functional patterns that indicate shared selective pressures.

Early comparative genomic studies on primates, mammals, bees, ants, *Drosophila* and other organisms (*Schlenke and Begun, 2003*; *Sackton et al., 2007*; *Kosiol et al., 2008*; *Barreiro and Quintana-Murci, 2010*; *Roux et al., 2014*) that included unbiased selection scans identified immune system pathways as common targets of natural selection. This implies that pathogens, which elicit the immune response, may be strong and consistent selective forces across species. Furthermore, in several clades of invertebrates, receptor genes, or the genes interacting directly with pathogens are most often the target of positive selection (*Sackton et al., 2007*; *Waterhouse et al., 2007*; *Ellis et al., 2012*). Finally, recent studies of mammals show that proteins that interact with viruses experience about twice as many amino acid changes compared to proteins that do not (*Enard et al., 2016*) and proteins that interact with *Plasmodium* experience elevated rates of adaptation (*Ebel et al., 2017*). While this evidence clearly implicates pathogens as a major selective force driving rapid evolution across a wide range of species, it is still unknown if pathogens tend to target a small, conserved set of host proteins (predicting repeated selection on the same genes), or

whether pathogens tend to interact with hosts in lineage or clade-specific ways (predicting selection on different genes). Previous genome-wide studies have focused on single taxonomic lineages (e.g. mammals), or limited subsets of candidate genes across clades, but the availability of many new genomes now allows detailed comparisons across deeply divergent clades to test the degree to which the specific genes represent shared hotspots of positive selection across long evolutionary timescales.

Birds (class Aves) are a natural genomic model to study shared selection pressures with mammals. Birds share a number of convergent features with mammals, including traits (e.g. homeothermy), and common pathogens (e.g. Influenza A). Furthermore, the number of bird genomes has increased dramatically in recent years (e.g. *Zhang et al., 2014a*). However to our knowledge, there have been no studies of genome-wide signatures of positive selection in this ecologically important group. Birds are a radiation of approximately 10,000 species (*Clements et al., 2016*) that possess diverse morphologies and behaviors (*Gill, 2007*). They have a global distribution and diverse range of habitats (*Jetz et al., 2012*), and many species migrate thousands of miles annually (*Gill, 2007*), making them excellent models for studies of disease ecology. From a genomic perspective, they have small genomes, generally stable chromosomes, a low number of repetitive elements relative to other vertebrates, and low rates of gene loss and gain (*Organ et al., 2007*; *Organ and Edwards, 2011*; *Zhang and Edwards, 2012*; *Ellegren, 2013*; *Zhang et al., 2014b*). Birds have the same general blueprint of immune pathways as mammals, but with a slimmed down gene repertoire and some small differences in the functions of specific genes (*Kaiser, 2010*; *Chen et al., 2013*; *Juul-Madsen et al., 2014*). Studies of the evolutionary dynamics of avian immune genes have almost exclusively focused on the major histocompatibility complex genes (MHC) or toll-like receptors (TLRs), with evidence of positive selection across species in MHC class I genes (*Alcaide et al., 2013*), MHC class II genes (*Edwards et al., 1995*; *Edwards et al., 2000*; *Hess and Edwards, 2002*; *Burri et al., 2008*; *Burri et al., 2010*) and TLRs (*Alcaide and Edwards, 2011*; *Grueber et al., 2014*; *Velová et al., 2018*). From a broader perspective, the conclusions drawn from more general studies of positive selection across birds have been limited by including only a few species (e.g. *Nam et al., 2010*), or using low-power analysis methods (e.g. comparing overall dN/dS values across GO-terms (*Zhang et al., 2014b*)).

We use comparative genomics in birds to study genome-wide signatures of positive selection without any *a priori* assumptions of gene functions. We find that the strongest signatures of selection are concentrated in four general categories: immune system genes, genes involved in recombination and replication, genes involved in lipid metabolism, and phototransduction genes. By comparing avian and mammalian datasets, we show that genes under positive selection in birds are likely to be under positive selection in mammals, and that this signal is the strongest in viral defense immune pathways. Finally, we reanalyzed 26 independent studies of gene expression after infectious challenge in birds and mammals as an unbiased, alternative approach to characterizing genes with potential immune function. We show that genes up-regulated following a pathogen challenge are more likely to be under positive selection in birds compared to those not differentially expressed, that there is also an overlap in birds and mammals in genes up-regulated in response to pathogens, particularly viruses, and that some of the classic genes studied as targets of host-pathogen co-evolution (PKR, MX1) are under selection and up-regulated following pathogen challenge in both clades. Together, all of our results support the hypothesis that adaptation against pathogens consistently involves the same genes across deep evolutionary timescales.

## Results

### Strong signatures of positive selection throughout the avian genome

We used PAML and HyPhy site models to test genes for evidence of positive selection across 39 species of birds (*Figure 1*; *Supplementary file 1* Table 1). We ran all models for 11,231 genes using the gene tree as the input tree and for 8669 genes using the species tree as the input tree. The 8669 genes used in the species tree analysis had a maximum of one sequence per species in the alignment, while the gene tree analysis included some alignments with up to three paralogs per species. To test for positive selection for each gene, we conducted likelihood ratio tests between models that include an extra ω parameter for some proportion of sites and models that do not include the

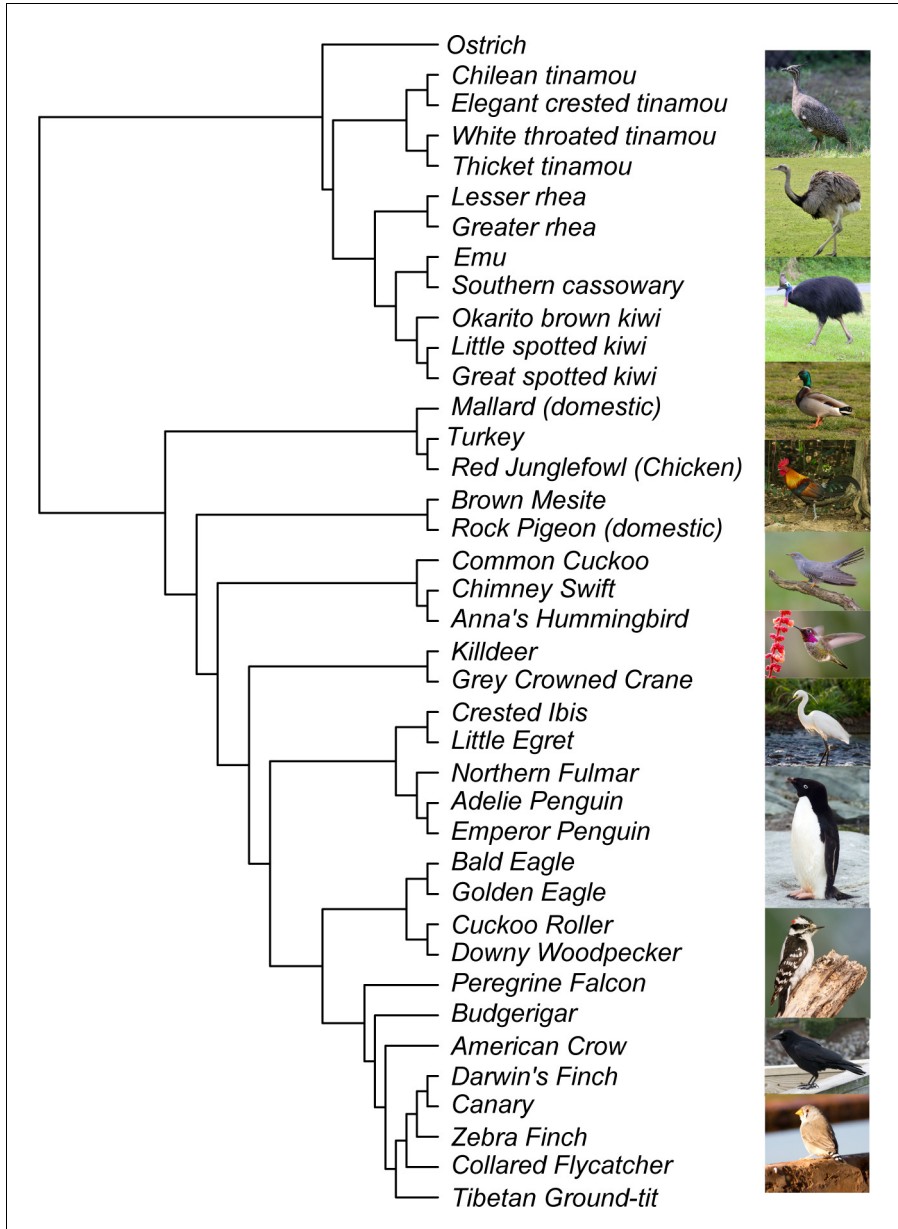

**Figure 1.** Species tree of 39 species used in study. Photo credit from Wikimedia Commons (top to bottom): Elegant Crested Tinamou: DickDaniels, Greater Rhea: Quartl, Southern Cassowary: Summerdrought, Mallard: Dcoetzee, Red Junglefowl: Francesco Veronesi, Common Cuckoo: Mike McKenzie, Anna's Hummingbird: Becky Matsubara, Little Egret: GDW.45, Adelie Penguin: Stan Shebs, Downy Woodpecker: Wolfgang Wander, American Crow: DickDaniels, Zebra Finch: Jim Bendon.
DOI: https://doi.org/10.7554/eLife.41815.002

extra ω parameter. An FDR-corrected p-value from that likelihood ratio test less than 0.05 is considered evidence of positive selection for that gene. For all model comparisons (PAML models described in *Table 1*), we found that between 17% and 73% of genes are under positive selection (*Table 2*). About 20% of genes are positively selected with the more conservative M1a vs. M2a tests or M2a vs M2a_fixed tests, with large overlaps among the genes identified. The less conservative M7 vs. M8 tests show much greater proportions of positively selected genes (~70%), although this is reduced to about 35% with the M8 vs. M8a test, indicating that the M8 model may often improve fit by adding a class of sites with ω very close to 1. HyPhy's BUSTED identified ~ 50% of genes as

**Table 1.** PAML Model descriptions.

| Model | Model description | Parameters |
|---|---|---|
| M0 | one ratio | $\omega$ |
| M1a | neutral | p0 (p1 = 1 p0) |
|  |  | $\omega 0 < 1$, $\omega 1 = 1$ |
| M2a_fixed | neutral | p0,p1 (p1 = 1 p0 - p1) |
|  |  | $\omega 0 < 1$, $\omega 1 = 1$, $\omega 2 = 1$ |
| M2a | selection | p0,p1 (p1 = 1 p0 - p1) |
|  |  | $\omega 0 < 1$, $\omega 1 = 1$, $\omega 2 > 1$ |
| M7 | neutral (beta distribution) | p, q |
| M8a | neutral (beta distribution) | p0 (p1 = 1 p0) |
|  |  | p, q, $\omega s = 1$ |
| M8 | selection (beta distribution) | p0 (p1 = 1 p0) |
|  |  | p, q, $\omega s > 1$ |

DOI: https://doi.org/10.7554/eLife.41815.003

positively selected (FDR-corrected p-value less than 0.05). Fewer than half of these genes are also identified as being positively selected by all PAML tests - 1562 genes with the gene tree as input and 1203 genes with species tree as input. In total, 14% of analyzed genes are found to be under positive selection in all tests (*Table 2*, *Supplementary file 1* Table 2 (raw gene tree results), *Supplementary file 1* Table 3 (raw species tree results)). We consider these 1562 genes to be a high-confidence positive selection gene set for downstream functional analyses. Compared to all other genes, the high-confidence positive selection gene set has overall higher distributions of M0 model $\omega$ values, which assumes a single $\omega$ for all sites in a gene (*Figure 2A*; Mann-Whitney U-test: gene trees: not-significant median $\omega$ = 0.084, significant median $\omega$ = 0.344, p<0.0001; species trees: not-significant $\omega$ = 0.084, significant $\omega$ = 0.332, p<0.0001).

Gene trees and species trees also had similar distributions of overall $\omega$ values with the M0 model (K-S test: D = 0.006, p=1, *Figure 2B*). The mean $\omega$ value is 0.15, the median $\omega$ is 0.10 and standard deviation is 0.14 using either the gene or species tree as the input tree. Because of the similarity between gene tree and species tree results, and to minimize issues associated with hemiplasy in species trees (*Hahn and Nakhleh, 2016*; *Mendes and Hahn, 2016*), for all bird-specific analyses below, we use focus on gene tree results to maximize the number of genes tested. However, all results are qualitatively similar with the species tree results as input.

We also explored whether alignment length had an effect on our ability to detect selection in a gene. We compared alignment lengths between our high-confidence genes detected as being under selection by all tests, and those that were not (*Figure 2C*). We assessed significance for this relationship using logistic regression, with selection status (under selection or not) as the dependent variable and alignment length as the independent variable. The model was significant for both gene trees (median length under selection = 1,818; median length not under selection = 1,089; coefficient = 3.49e-4, p-value<0.0001) and species trees (median length under selection = 1,893; median length not under selection = 1,158; coefficient = 3.46e-4, p-value<0.0001), likely indicating that power to detect selection increases with alignment length.

**Table 2.** Counts (above) and proportions (*below*) for all tests of individual, and combined tests of selection for gene trees and species trees.

| Dataset | N genes | m1a vs m2a | m2a vs m2a_fixed | m7 vs m8 | m8 vs m8a | All PAML | Busted | All PAML + BUSTED |
|---|---|---|---|---|---|---|---|---|
| Gene trees | 11231 | 1925 | 2197 | 7504 | 3679 | 1901 | 6244 | 1562 |
|  |  | *0.17* | *0.20* | *0.67* | *0.33* | *0.17* | *0.56* | *0.14* |
| Species trees | 8669 | 1783 | 2026 | 6293 | 3395 | 1752 | 3870 | 1203 |
|  |  | *0.21* | *0.23* | *0.73* | *0.39* | *0.20* | *0.45* | *0.14* |

DOI: https://doi.org/10.7554/eLife.41815.004

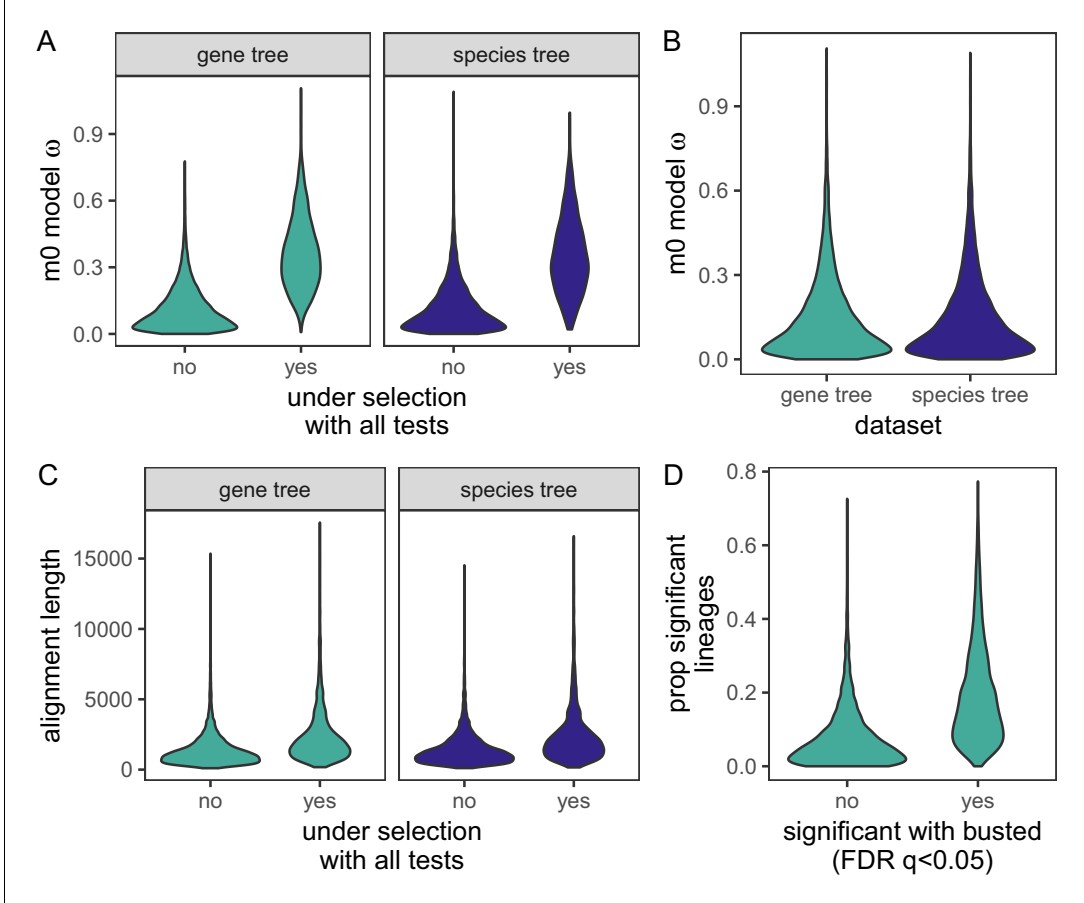

**Figure 2.** Distributions of gene-wide statistics using gene trees or species trees as the input phylogeny. (A) Comparison of M0 model ω values between genes under selection for all site tests, and those not under selection, for both gene trees and species trees. The mean ω values were significantly higher for genes under selection for both gene trees and species trees (Mann-Whitney U-test: gene trees: W = 1201387, p < 0.0001; species trees: W = 938934, p < 0.0001). (B) Violin plot of ω values from the PAML M0 model using either gene trees or species trees as the input phylogeny. (C) Violin plot of alignment lengths for genes significant in all tests of selection, or not significant in all tests of selection using either gene trees or species trees as input phylogenies. (D) Distribution of the proportion of significant lineages for HOGs identified as not significant (FDR-corrected p-value ≥ 0.05), or significant (FDR-corrected p-values < 0.05) with BUSTED. The means of the two distributions are significantly different (Mann-Whitney U-test: W= 6205530, p<10-16).

DOI: https://doi.org/10.7554/eLife.41815.005

## Immune, recombination, lipid metabolism, and phototransduction pathways are enriched for positive selection in birds

For an unbiased perspective on whether or not positively selected genes are concentrated in particular functional pathways, we performed a pathway enrichment test of positively selected genes against a background of all genes tested. With chicken as the reference organism, 351 genes of the high-confidence positive selection gene set and 3347 of all genes tested could be mapped to KEGG pathways for use as the test set and gene universe respectively. Out of the 166 KEGG pathways available for chicken, or any other bird species, 119 had at least one gene with evidence of positive selection (*Supplementary file 1* Table 4). We found 18 KEGG pathways that were significantly enriched with positively selected genes (q-value less than 0.1; *Figure 3A*; *Supplementary file 1* Table 4). These 18 pathways belong to seven KEGG functional categories: infectious disease, immune system, signaling molecules and interaction, replication and repair, lipid metabolism, and sensory system. Some genes are shared by multiple enriched pathways, particularly among those with immune or recombination-related functions. However, many genes are uniquely enriched in a single pathway as well (*Figure 3B*), suggesting that these enrichment results are not driven by a few

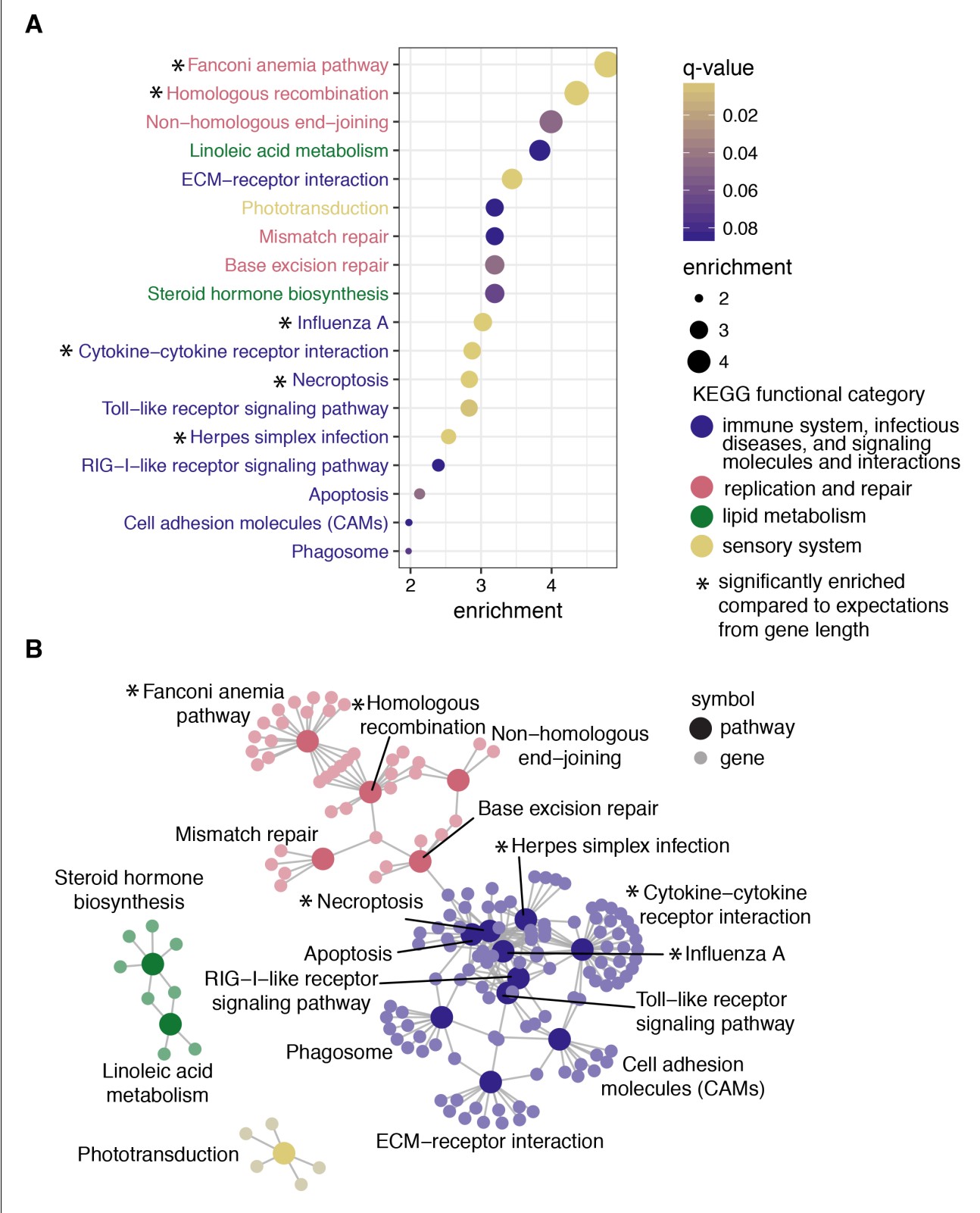

**Figure 3.** Pathway enrichment results to determine whether positively selected genes are functionally similar with chicken as the reference organism. (A) The 18 pathways significant at q-value < 0.1 ordered by enrichment values, calculated as the proportion of genes under selection in the pathway over the proportion of genes significant in all KEGG pathways. Points are filled by q-value, and each pathway is colored by the broader KEGG functional category. Edge lengths do not contain meaningful values, but are chosen to maximize viewability. Pathways with more genes under selection than

*Figure 3 continued on next page*

*Figure 3 continued*

expected based on median gene length are highlighted with an asterisk. (**B**) Map depicting the relationships of all significant genes among pathways. Each gene (small, light circle) is connected by a line to each pathway it belongs to (large, dark circle). Each point is shaded according to the broader KEGG functional category. See *Figure 3—figure supplement 1* for pathways identified using species trees.

DOI: https://doi.org/10.7554/eLife.41815.006

The following figure supplement is available for figure 3:

**Figure supplement 1.** Pathways enriched using results from species trees as the input phylogeny.

DOI: https://doi.org/10.7554/eLife.41815.007

core genes present in many pathways. While we focus on the gene tree results due to the larger number of genes available for enrichment tests, species tree results show the same overall trends of significant immune and recombination-related pathways (*Figure 3—figure supplement 1*).

To test whether our pathway enrichment results were robust to reference organism, we also conducted pathway enrichment tests using zebra finch and human as the reference organism. Pathway enrichment results using zebra finch showed similar results as those presented above using chicken, particularly for immune-related pathways and recombination and repair pathways (*Supplementary file 1* Table 5). Pathway enrichment results using human, which has additional annotated pathways, resulted in 37 pathways significantly enriched with positively selected genes (q-value less than 0.1; *Supplementary file 1* Table 6) out of 269 pathways with at least one homologous gene in our dataset. Compared to the enrichment results using chicken, the human pathways primarily added many disease or immune pathways not available for birds, suggesting that the overall functional results are robust to reference organism.

Finally, to test whether gene length variance among KEGG pathways influenced our results, we estimated the expected number of genes under selection among genes annotated to a given pathway using the parameters from our logistic regression model conducted for all genes (see previous section), and compared these values to our observed results. We used a conservative Fisher's exact test to test for significance, and find that while all of the pathways we observed have more genes under selection than expected based on median gene length alone (*Figure 3*; *Supplementary file 1* Table 7), not all are statistically significant.

## Lineages clustered by genes under selection in birds are most strongly related to body size and lifespan

Codon-based site models typically can only detect positive selection when the same sites in the protein are under selection in numerous lineages. In order to detect selection limited to particular lineages, we relaxed this assumption, and used aBS-REL to estimate of the probability of selection independently at each branch of the phylogeny. To test consistency with our site-model results, we calculated the number of lineages with evidence for positive selection from the branch-site (aBS-REL) for each gene. We found that genes identified by BUSTED as having sites with evidence of positive selection across avian lineages also had significantly more lineages under selection using aBS-REL (Mann-Whitney U-test: median proportion significant lineages under selection given significant BUSTED result: 0.16, median proportion significant lineages under selection given non-significant BUSTED result: 0.05, $p<10^{-16}$, *Figure 2D*). While branch-site tests can be subject to false inferences of positive selection due to multinucleotide mutations (*Venkat et al., 2018*), the similar patterns between our site-model results, and the aBS-REL results suggests that the overall patterns we observe hold true with this alternative analysis.

To identify additional functional classes of genes that may be selected in only a subset of lineages, we used a principle components analysis (PCA) to summarize the variance of parameter estimates (proportion of sites under selection, ω) across genes for each species, and then used phylogenetic comparative methods to identify species traits associated with PC loadings for each gene. We find three PCs that together explain 16.8% to 17.2% of the variance in aBS-REL parameter estimates across species, while the remaining PCs explain about 3% of the variance (*Figure 4—figure supplement 4*). PC1 (*Figure 4*) identifies a diverse set of species from different lineages sharing similar PC scores, while PC2 and PC3 appear to primarily identify lineage or clade-specific selection with variation depending on parameter and underlying tree used (gene or species trees). We

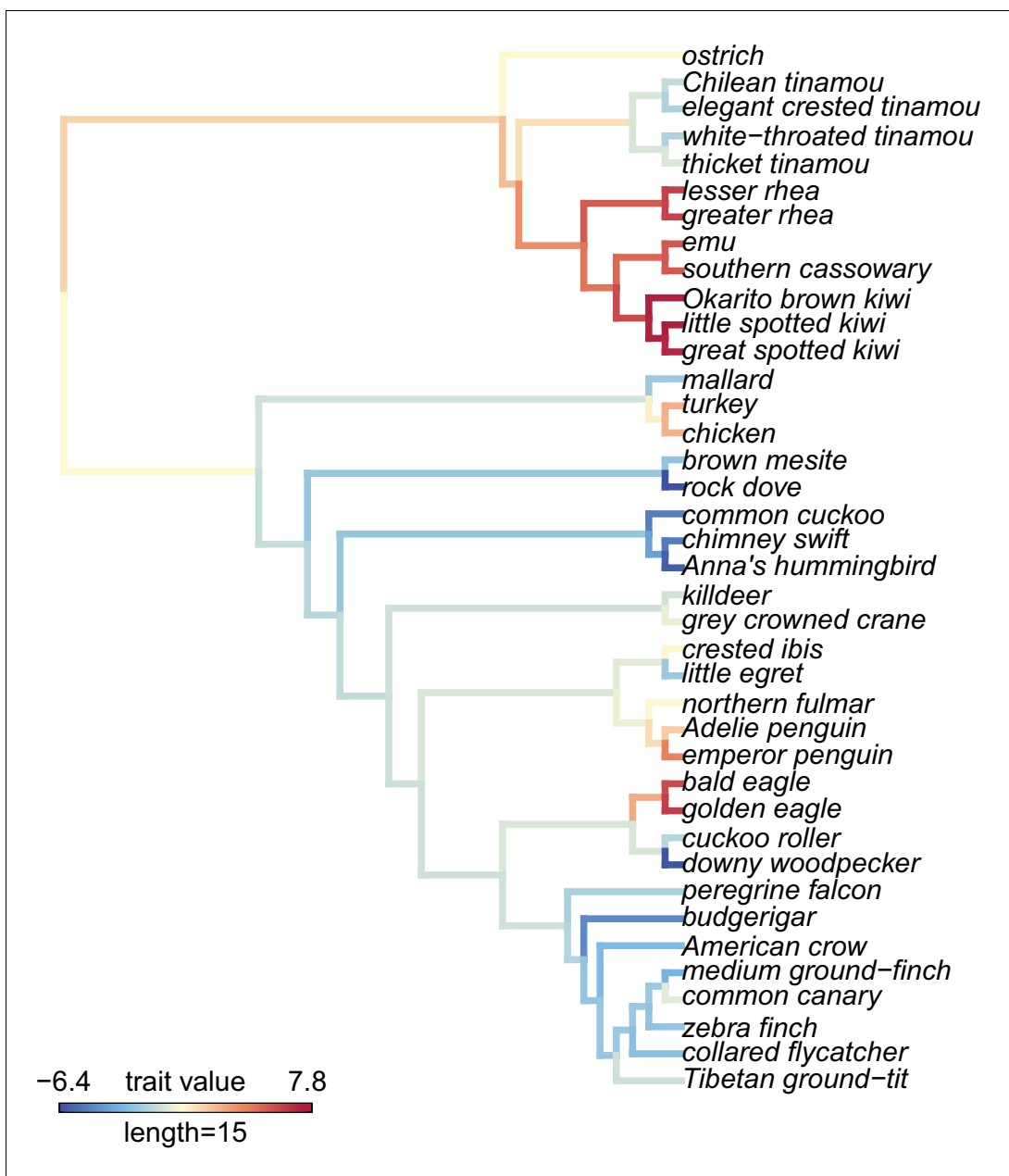

**Figure 4.** A visualization of PC1 scores estimated by the proportion of sites under selection using gene trees on the phylogeny and the maximum likelihood reconstruction of the PC1 values for internal branches. The PC1 scores indicate species in different clades that have similar parameter estimates from aBS-REL tests for positive selection and explain 7.2% of the variance across all genes tested. See Figure 4 supplement 1 for a visualization of PC1 estimated using ω values and gene trees, Figure 4 supplement 2 for a visualization of PC1 estimated using the proportion of sites under selection using species trees, Figure 4 supplement 3 for a visualization of PC1 estimated using ω using species trees, and Figure 4 supplement 4 for a visualization of eigenvalues for all PCAs.

DOI: https://doi.org/10.7554/eLife.41815.008

The following figure supplements are available for figure 4:

**Figure supplement 1.** (A) visualization of PC1 scores estimated by the ω estimates using gene trees on the phylogeny and the maximum likelihood reconstruction of the PC1 values for internal branches.

DOI: https://doi.org/10.7554/eLife.41815.009

**Figure supplement 2.** (A) visualization of PC1 scores estimated by the proportion of sites under selection using species trees on the phylogeny and the maximum likelihood reconstruction of the PC1 values for internal branches.

*Figure 4 continued on next page*

*Figure 4 continued*

DOI: https://doi.org/10.7554/eLife.41815.010

**Figure supplement 3.** (A) visualization of PC1 scores estimated by the ω estimates using species trees on the phylogeny and the maximum likelihood reconstruction of the PC1 values for internal branches.

DOI: https://doi.org/10.7554/eLife.41815.011

**Figure supplement 4.** Visualization of the variance explained by the first 10 PC axes (scree plot).

DOI: https://doi.org/10.7554/eLife.41815.012

conducted subsequent analyses using gene trees as input, given the similarity in results when using gene trees or species trees as the baseline phylogeny, so that we could maximize the number of genes available for analysis. We focused on PC1, the only principle component with similar values in unrelated lineages that varied consistently between parameters (*Figure 4*) to test whether it might be associated with life history. We found a significant correlation between log-transformed body mass, a proxy for many life history characteristics, and PC1 scores for each species using phylogenetic generalized least squares (*Figure 5*, proportion of sites under selection: β = −1.04, SE = 0.24, t-value = −4.26, p-value=0.0001; ω: β = −10.24, SE = 3.01, t-value = −3.40, p-value=0.0016).

To understand any functional signal in the genes most strongly correlated with body size, we calculated a Spearman's rank correlation for each gene using the proportion of sites under selection or log-transformed aBS-REL ω values for each lineage compared to log-transformed body mass. We performed KEGG pathway gene set enrichment using the ρ value for each gene. For the proportion of sites under selection, the most significant pathway was the cellular senescence pathway, which had a q-value of 0.61 and a normalized enrichment score of −1.52. For ω, the most significant pathway was the FoxO signaling pathway, which had a q-value of 0.09 and a normalized enrichment score of −0.62 (*Supplementary file 1* Table 8).

## Shared signatures of selection in birds and mammals are enriched for viral-interacting pathways

We investigated whether we could detect signatures of pathogen-mediated selection at deeper time scales by testing whether the same genes are repeatedly under selection in both birds and mammals, and whether those genes are clustered in functional pathways. We combined our results with those from *Enard et al., 2016*, a study that used HyPhy's BUSTED program to test for positive selection in 9681 orthologous genes from 24 mammal genomes. To best match the experimental procedures used by *Enard et al., 2016*, for bird-mammal comparisons we used only our BUSTED results with the species tree as the input phylogeny. The combined dataset consisted of 4931 orthologous genes with results in both clades.

We first tested for significant overlap in positively selected genes in both clades with a Fisher's exact test. To understand whether these results were driven by genes with different levels of evidence for positive selection, we used four different FDR-corrected p-value cutoffs for significance, 0.1, 0.01, 0.001, and 0.0001. We found evidence for a significant overall overlap in positively-selected genes at all four different FDR-corrected p-value cutoffs, with stronger signal at smaller FDR-corrected p-values (*Figure 6A*, *Supplementary file 1* Table 9). To ensure that these results were not driven by greater power to detect selection in more constrained genes, or longer genes, we conducted two additional analyses. First, we performed the Fisher's exact test after removing the 20% most constrained genes, defined as the 20% of genes with the smallest m0 ω values. The results of this test were extremely similar to the previous test (*Figure 6—figure supplement 1*). Second, we conducted logistic regressions at each FDR-corrected p-value cutoff with the selection status in mammals as the response variable, and the selection status in birds, alignment length, and their interaction as the independent variables. At all FDR-corrected p-value cutoffs, both alignment length and selection status in birds are significant predictors of selection status in mammals (*Supplementary file 1* Table 10), but their interaction is not. These results suggest that the overlap we observe between birds and mammals cannot be explained by either constraint bias or alignment length, and is instead likely related to the biology of the shared selected genes.

We tested for functional enrichment in shared selected genes compared to all genes under selection in birds. KEGG enrichment with a test set of genes under selection in both clades and gene universe of genes under selection in birds, showed that pathways with immune function, particularly

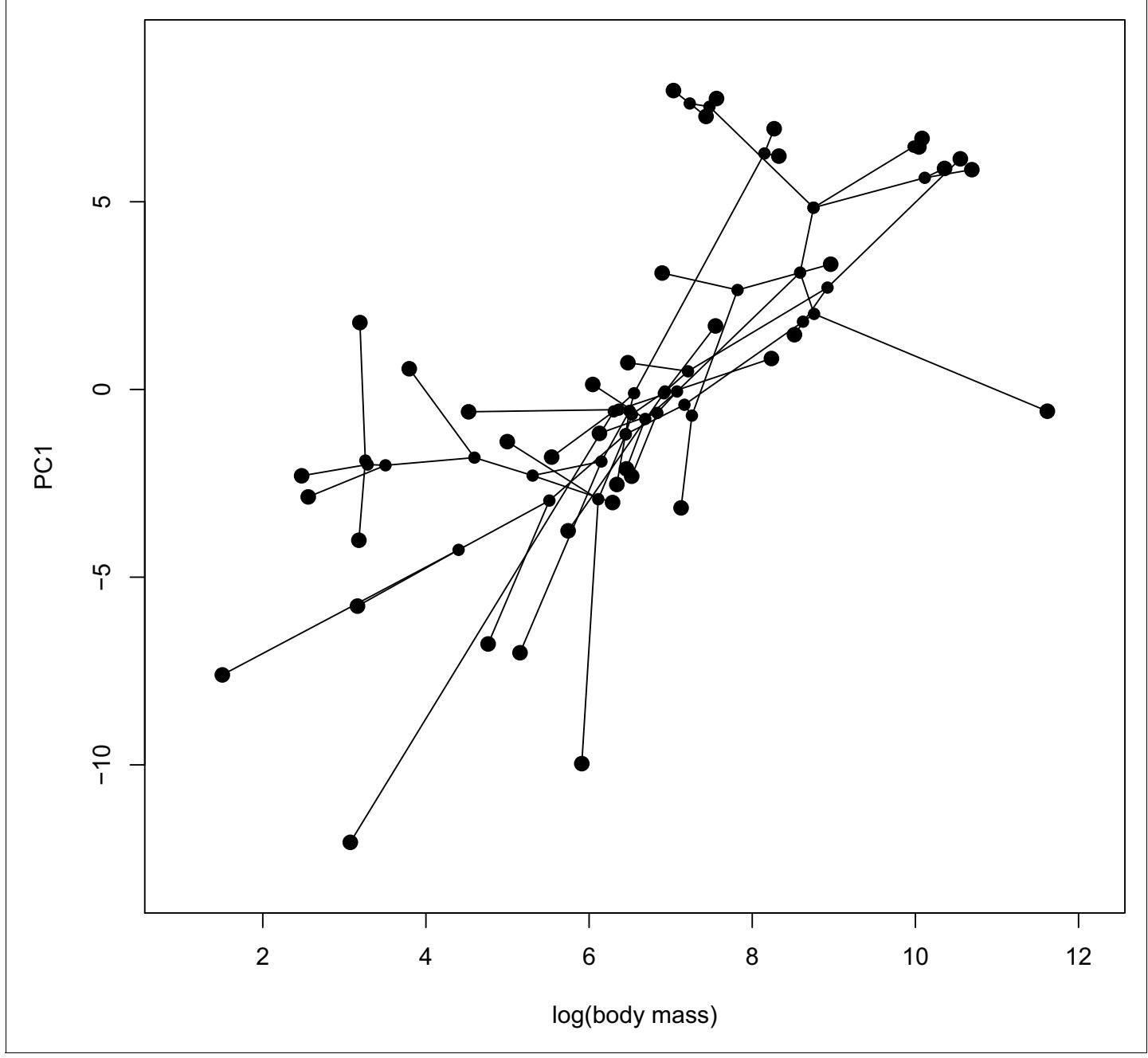

**Figure 5.** A phylomorphospace plot showing the association between log-transformed body mass on the x-axis and PC1 estimated from the proportion of sites under selection on the y-axis. Species values and reconstructed node values are connected by phylogeny. A PGLS analysis of these two traits showed a significant correlation (p=0.0001). See *Figure 5—figure supplement 1* for a phylomorphospace plot depicting PC1 estimated using ω estimates.

DOI: https://doi.org/10.7554/eLife.41815.013

The following figure supplement is available for figure 5:

**Figure supplement 1.** A phylomorphospace plot showing the association between log-transformed body mass on the x-axis and PC1 estimated from the ω estimates on the y-axis.

DOI: https://doi.org/10.7554/eLife.41815.014

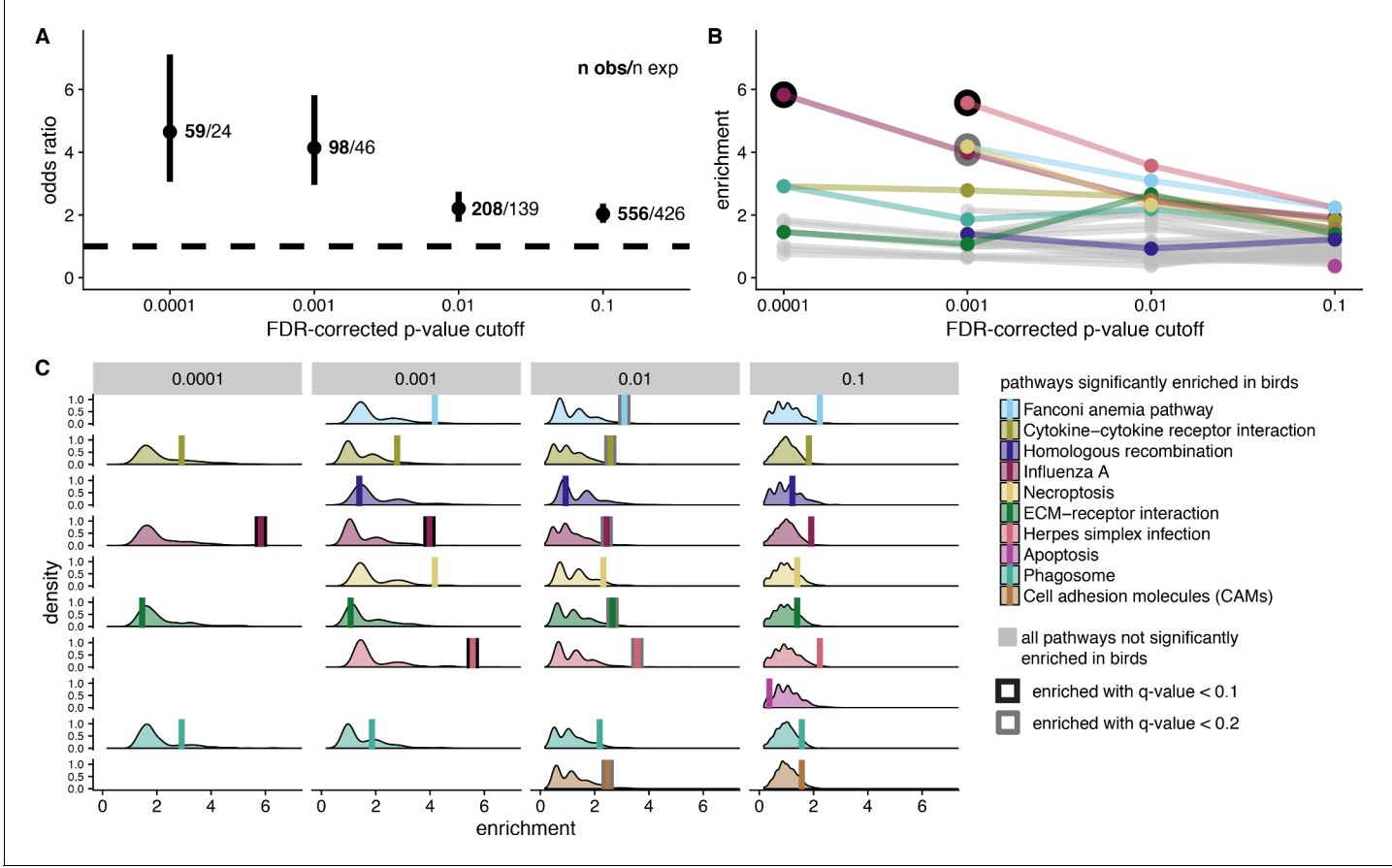

**Figure 6.** Signatures of shared positive selection in birds and mammals. For all analyses, we considered four different FDR-corrected p-value cutoffs for significance (to identify genes under positive selection) A. Odds ratio of overlap in genes under selection in both bird and mammal datasets. We indicate the number of observed genes under selection in both clades (n obs) and the number of expected genes under selection in both clades (n exp). B. Pathway enrichment scores from KEGG pathway enrichment tests with genes under selection in both birds and mammals as the test set, and genes under selection in birds as the background set. Ten pathways significantly enriched in birds with at least one gene under selection in both birds and mammals are color-coded. All other pathways are shown in grey. Significant enrichment values are outlined in black (q-value < 0.1) or grey (q-value < 0.2). C. Null distribution of enrichment scores generated from 1,000 randomization tests compared to empirical enrichment scores (vertical bars). Null distributions were generated by randomly selecting gene sets from the background set of genes (bird significant genes) for use as the test set. The randomized test set contained the same number of genes as empirical test set for each FDR-corrected p-value cutoff for significance. Empirical enrichment scores are depicted by a vertical bar, and with significant q-value scores outlined in black (q-value < 0.1) or grey (q-value < 0.2). See Figure 6 Supplement 1 for the odds ratio overlap in genes under selection in both bird and mammal datasets with the 20% most constrained genes removed.
DOI: https://doi.org/10.7554/eLife.41815.015

The following figure supplement is available for figure 6:

**Figure supplement 1.** Odds ratio overlap of bird and mammal genes under selection with 20% most constrained genes (as estimated by the m0 model) removed.
DOI: https://doi.org/10.7554/eLife.41815.016

viral-interacting pathways, are significantly enriched for shared signatures of selection. These results are particularly significant at the lowest FDR-corrected p-value significance cutoffs (*Figure 6B*). As pathways enriched for positively selected genes in birds have higher enrichment values than other pathways (*Figure 6B*), we also conducted 1000 randomized enrichment tests to make sure pathways with more genes under selection in birds are not more likely to show more positively selected genes in both lineages by chance. We calculated multiple test corrected p-values for the empirical enrichment scores compared to the randomly generated null distribution within each of the four FDR-corrected p-value cutoffs for significance. These results corroborate those of KEGG enrichment tests, with Influenza A and Herpes simplex infection pathways showing significantly higher enrichment values, particularly at lower FDR-corrected p-value cutoffs for significance (*Figure 6C*). We observe four

genes under selection in the Influenza A pathway out of 12 genes annotated to the pathway, compared to an expectation of 0.7 using an FDR-corrected p-value cutoff of 0.0001, and four genes under selection in both Influenza A and Herpes simplex pathways compared to expectations of 1 and 0.7, respectively, out of 14 and 10 pathway genes using an FDR-corrected p-value cutoff of 0.001.

## Genes up regulated in response to pathogens are more likely to be under positive selection in birds

We used gene expression data as an alternative test of whether pathogens are likely driving immune-related patterns of positive selection. First, for birds, we tested whether genes that were significantly differentially expressed following a pathogen challenge are more likely to be under positive selection. We used avian transcriptome data from 12 different studies representing seven different types of pathogens, including four viruses, two bacteria, and one species of protist (*Supplementary file 1* Table 11). We compared both the proportions of positively-selected genes (using the high-confidence positive selection gene set) that were up-regulated following a pathogen challenge compared to those not differentially expressed, and the proportions of positively selected genes that were down-regulated following a pathogen challenge compared to those not differentially expressed. We found that for all pathogens, up-regulated genes are significantly more likely to be positively selected (*Figure 7*; *Supplementary file 1* Table 12). The pattern was less consistent for down-regulated genes, with overall smaller numbers of down-regulated genes, weak evidence for a greater proportion of positively selected genes for West Nile Virus and *Plasmodium*, and weak evidence for a smaller proportion of positively selected genes for *E. coli* (*Figure 7*; *Supplementary file 1* Table 12). Note that we did not test whether our positively selected genes were under selection in the specific lineages where the transcriptome data were collected. Therefore, we cannot say for sure that there is a direct link between selection and expression in each specific dataset. However, the observed correlation between gene expression and selection status only strengthens our hypothesis that these genes are under long-term co-evolutionary relationships between pathogens and hosts.

We tested for shared pathogen-mediated selection in birds and mammals by comparing bird and mammal gene expression patterns when challenged with the same, or a closely-related pathogen. There were five pathogens with publicly available data for at least one bird and mammal species – two viruses: Influenza A and West Nile virus, two bacteria: *E. coli* and mycoplasma, and one protist: *Plasmodium* (*Supplementary file 1* Table 11). All comparisons between bird and mammal datasets for viral and bacterial pathogens showed that there was significant overlap in up-regulated genes, but no significant overlap in down-regulated genes (*Table 3*). Genes differentially expressed in response to *Plasmodium* showed the opposite pattern, with significant overlap in down-regulated genes, but no significant overlap in up-regulated genes (*Table 3*).

Logistic regressions with genes under selection in birds as the response variable and genes under selection in mammals, genes up or down-regulated in birds, and their interaction as predictor variables showed that for all categories, the selection status in mammals is the strongest predictor, followed by the transcriptional response in birds for some pathogens, but no significant interaction between the two (*Table 4*). Very few genes were under selection in birds and mammals and also differentially expressed in both clades at all FDR-

corrected p-value cutoffs for significance, reducing power to detect signals of enrichment. However, a few genes with low FDR-corrected p-values selection cutoffs in both datasets (p<0.0001) were also up-regulated in response to influenza (PKR, PARP9, and MX1), up-regulated in response to West Nile virus (PKR), up-regulated in response to *E. coli* (F5), or down-regulated in response to *E. coli* (RAD9A).

Due to the small number of genes under selection and differentially expressed in both lineages, we also sought to test whether there was any difference in differential expression effect size (β values) between genes under selection in both lineages, genes under selection in birds, and genes not under positive selection. A difference in overall differential expression effect size might suggest the existence of general differences in gene expression patterns that might not be strong enough to produce significant signal at individual genes. For each gene, we calculated the harmonic mean of bird and mammal absolute, standardized β values in response to infection with each pathogen and compared the mean of each β distribution in the three selection categories with pairwise Mann-

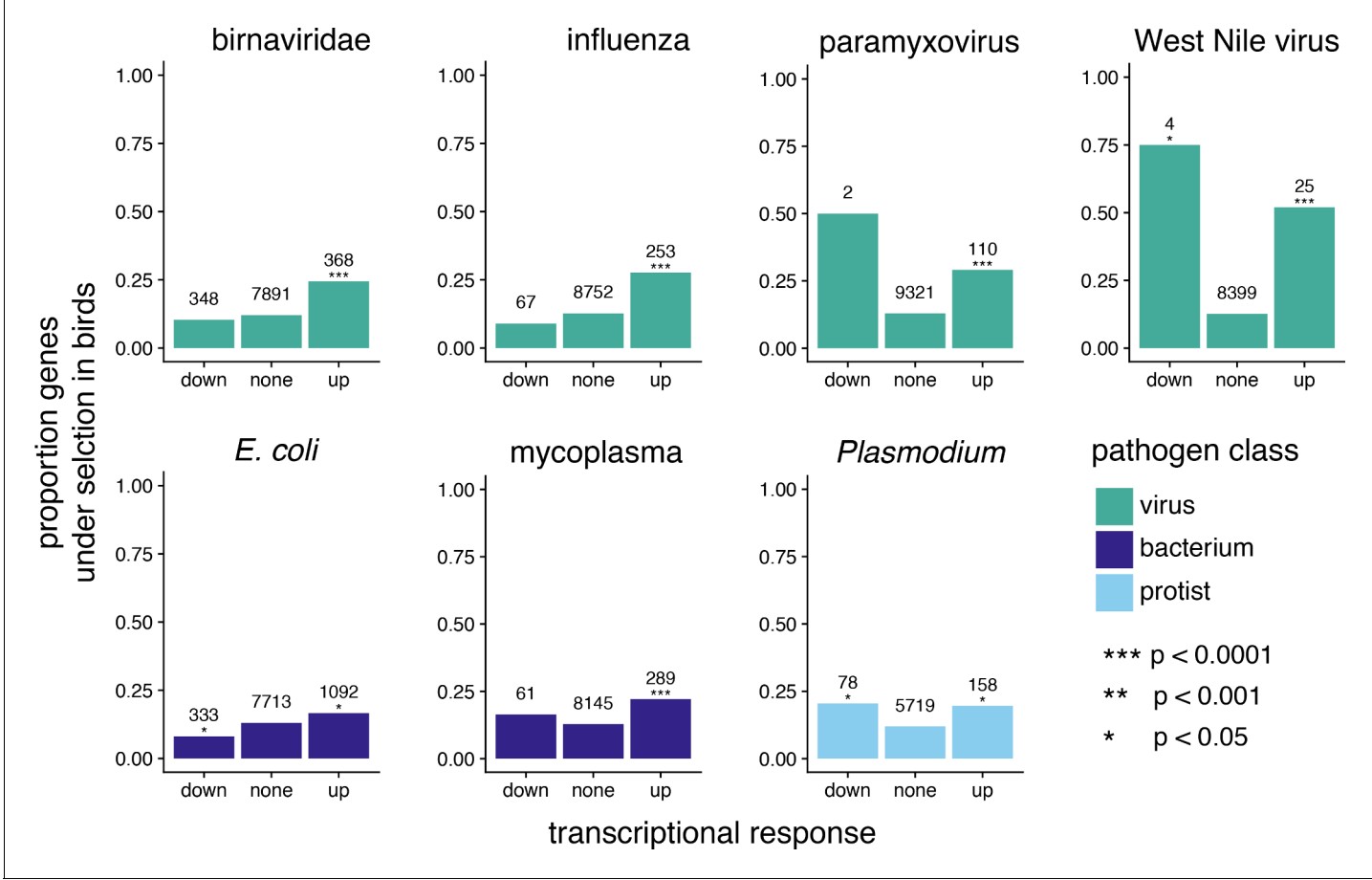

**Figure 7.** A comparison of genes under positive selection in birds and genes differentially expressed following pathogen challenge to test for patterns of pathogen-mediate selection. For different pathogens, we show the proportion of genes under positive selection in birds (defined as significant with FDR corrected p-value < 0.05 for all PAML and BUSTED model comparisons) for genes down significantly down regulated, significantly up regulated, or not significantly differentially regulated. The number above each bar indicates the number of genes in a given transcriptional response class. The significance of enrichment for positively-selected genes in up- or down-regulated expression classes, as calculated by logistic regression, is indicated by asterisks above the "down" and "up" bars.

DOI: https://doi.org/10.7554/eLife.41815.017

Whitney U-tests. We found that genes under selection in both lineages have larger β values than both other classes, particularly in response to viruses (**Figure 8**, **Supplementary file 1** Table 13). Genes under selection in birds also have larger β values compared to genes not under selection in response to viruses, but not other pathogens.

## Discussion

Here, we show that shared signatures of positive selection are consistent with pathogen-mediated selection. First, across birds, genes involved in immune system function, DNA replication and repair, lipid metabolism, and phototransduction are targets of positive selection. Most of these pathways can be directly or indirectly linked to immune response. Functional transcriptomic data independently validate these results, showing that gene up-regulated in response to pathogens contain a higher proportion of genes under positive selection than those not regulated by infection. These results hold true at a broader taxonomic scale. We not only show that genes under selection in birds are more likely to be under selection in mammals, but that these shared selected genes are enriched for immune system processes, and in particular those related to viral response. We find few genes differentially regulated and under positive selection in both birds and mammals, but those we found

**Table 3.** Fisher's exact test results from bird and mammal transcriptome studies.

| Pathogen | Transcriptional response | N genes diff. expressed in both lineages | N genes expected by chance | p-value | Odds ratio (95% conf. intervals) |
|---|---|---|---|---|---|
| Influenza | up | 30 | 14.8 | <0.0001 | 2.48 (1.56,3.84) |
| Influenza | down | 7 | 7.2 | 1.000 | 0.96 (0.35,2.29) |
| West Nile Virus | up | 6 | 0.9 | <0.0001 | 25.06 (4.46,253.47) |
| West Nile Virus | down | 0 | 0 | 1.000 | 0 (0, 884.62) |
| *E. coli* | up | 84 | 52.4 | <0.0001 | 1.9 (1.45, 2.47) |
| *E. coli* | down | 24 | 20.7 | 0.397 | 1.2 (0.73, 1.90) |
| Mycoplasma | up | 16 | 8.4 | 0.010 | 2.1 (1.14, 3.62) |
| Mycoplasma | down | 0 | 0.1 | 1.000 | 0 (0, 47.56) |
| Plasmodium | up | 14 | 9.8 | 0.166 | 1.53 (0.79, 2.77) |
| Plasmodium | down | 9 | 3.2 | 0.004 | 3.44 (1.42, 7.57) |

DOI: https://doi.org/10.7554/eLife.41815.018

are known to interact directly with pathogens. We also find that genes under positive selection in both lineages have significantly larger overall differential expression effect values compared to those only under positive selection in birds or those not under positive selection. Our results point to pathogens, and in particular viruses, being the most consistent selective pressure across tetrapods.

## Pathogens drive shared signatures of selection across birds and mammals

The strong overlap in genes under positive selection in mammals and birds (*Figure 6A*), together with the functional enrichment and expression results, support the hypothesis that pathogens consistently target the same genes across deeper evolutionary timescales. Although there are some differences in the fine details between avian and mammalian immune systems (e.g. different TLRs are functionally similar in the types of pathogens they recognize (*Kaiser, 2010*; *Chen et al., 2013*), the overall immune responses are conserved between the clades (*Kaiser, 2010*). *Schrom et al., 2018* theoretically demonstrated that there are a limited number of network architecture configurations that are both inducible and robust, and our results here further suggest that pathogens are constrained in how they can interact with these networks to suppress an immune response. Further work on signatures of selection in other tetrapod clades would help to distinguish whether the shared patterns of selection we observe are due to convergence or ancient shared tetrapod selection.

We also show that the same genes are likely to be up-regulated in response to pathogen infection (*Table 3*), despite significant differences in the overall transcriptomic study designs (*Supplementary file 1* Table 11). We found few genes both differentially expressed and under positive selection in both lineages, although we did find a significant, but small tendency for genes under selection in both clades to be differentially expressed (*Figure 8*). Those genes we did find are either classic examples of well-documented host-pathogen arms races (PKR (*Samuel et al., 2006*; *Rothenburg et al., 2009*; *Elde et al., 2009*; *Enard et al., 2016*) and MX1 (*Ferris et al., 2013*)), or genes known to interact directly with pathogens or the immune response (PARP9 (*Zhang et al., 2015*), RAD9 (*An et al., 2010*), F5 (*Brunder et al., 1997*)), which are new candidates for genes that may be involved in host-pathogen arms races across tetrapods.

## Viruses produce the strongest signatures of pathogen-mediated selection

Our results highlight that shared signatures of selection are enriched for pathways annotated to interact with viruses compared to those that interact with other pathogens. We show similar findings with our differential expression results - the differences in shared levels of differential expression in birds and mammals are strongly significant for the viral infectious agents, and only marginally

significant for the other infectious agents (*Figure 8*). There was also a stronger overlap in expressed genes for the viruses compared to the other two classes of pathogens (*Table 3*). Finally, the Influenza A and Herpes simplex pathways were significantly enriched for shared genes under selection (*Figure 6B,C*). There is a near universal tendency to switch hosts in viruses (*Geoghegan et al., 2017*; *Shi et al., 2018*) and retroviruses (*Henzy et al., 2014*), although there is some variation in the prevalence of host-switching in different viral families (*Geoghegan et al., 2017*). This tendency may drive shared selection pressures if viruses target common host genes across divergent species. Examples of host-switching will only increase as more viruses are sequenced. Across populations of *Drosophila melanogaster*, a recent study observed higher rates of adaptation in viral genes only, not immune genes in general, bacterial genes, or fungal genes (*Early et al., 2017*), suggesting that these results may be more general across broader organisms as well.

**Table 4.** Logistic regression results testing whether genes under selection in birds could be predicted by selection status in mammals (sig_mammals), transcriptional regulation in birds, or their interaction.

| Pathogen | Transcriptional response | N genes | Predictor variable | Estimate | Standard error | Z score | p-value |
|---|---|---|---|---|---|---|---|
| Influenza | down | 4488 | sig_mammals | 0.76 | 0.09 | 8.45 | <0.0001 |
| Influenza | down | 4488 | down_reg_birds | −0.12 | 0.42 | −0.29 | 0.771 |
| Influenza | down | 4488 | sig_mammals: down_reg_birds | −0.76 | 0.85 | −0.89 | 0.372 |
| Influenza | up | 4488 | sig_mammals | 0.77 | 0.09 | 8.52 | <0.0001 |
| Influenza | up | 4488 | up_reg_birds | 0.79 | 0.22 | 3.69 | 0.0002 |
| Influenza | up | 4488 | sig_mammals: up_reg_birds | −0.88 | 0.5 | −1.77 | 0.077 |
| West nile virus | down | 3774 | sig_mammals | 0.77 | 0.1 | 7.76 | <0.0001 |
| West nile virus | down | 3774 | down_reg_birds | 11.98 | 196.97 | 0.06 | 0.952 |
| West nile virus | down | 3774 | sig_mammals: down_reg_birds | - | - | - | - |
| West nile virus | up | 3774 | sig_mammals | 0.77 | 0.1 | 7.76 | <0.0001 |
| West nile virus | up | 3774 | up_reg_birds | 1.1 | 0.87 | 1.27 | 0.203 |
| West nile virus | up | 3774 | sig_mammals: up_reg_birds | −1.46 | 1.66 | −0.88 | 0.378 |
| *E. coli* | down | 4225 | sig_mammals | 0.74 | 0.09 | 7.95 | <0.0001 |
| *E. coli* | down | 4225 | down_reg_birds | −0.16 | 0.19 | −0.83 | 0.409 |
| *E. coli* | down | 4225 | sig_mammals: down_reg_birds | 0.18 | 0.5 | 0.36 | 0.717 |
| *E. coli* | up | 4225 | sig_mammals | 0.72 | 0.1 | 7.2 | <0.0001 |
| *E. coli* | up | 4225 | up_reg_birds | 0.24 | 0.1 | 2.39 | 0.017 |
| *E. coli* | up | 4225 | sig_mammals: up_reg_birds | −0.03 | 0.26 | −0.13 | 0.895 |
| Mycoplasma | down | 4059 | sig_mammals | 0.73 | 0.09 | 7.74 | <0.0001 |
| Mycoplasma | down | 4059 | down_reg_birds | −0.59 | 0.53 | −1.11 | 0.266 |
| Mycoplasma | down | 4059 | sig_mammals: down_reg_birds | −0.06 | 0.93 | −0.06 | 0.948 |
| Mycoplasma | up | 4059 | sig_mammals | 0.75 | 0.1 | 7.77 | <0.0001 |
| Mycoplasma | up | 4059 | up_reg_birds | 0.51 | 0.21 | 2.42 | 0.016 |
| Mycoplasma | up | 4059 | sig_mammals: up_reg_birds | −0.71 | 0.41 | −1.73 | 0.084 |
| Plasmodium | down | 3222 | sig_mammals | 0.74 | 0.11 | 6.86 | <0.0001 |
| Plasmodium | down | 3222 | down_reg_birds | −0.02 | 0.37 | −0.05 | 0.961 |
| Plasmodium | down | 3222 | sig_mammals: down_reg_birds | 0.01 | 0.85 | 0.01 | 0.992 |
| Plasmodium | up | 3222 | sig_mammals | 0.73 | 0.11 | 6.71 | <0.0001 |
| Plasmodium | up | 3222 | up_reg_birds | 0.19 | 0.23 | 0.85 | 0.396 |
| Plasmodium | up | 3222 | sig_mammals: up_reg_birds | 0.44 | 0.87 | 0.5 | 0.614 |

DOI: https://doi.org/10.7554/eLife.41815.019

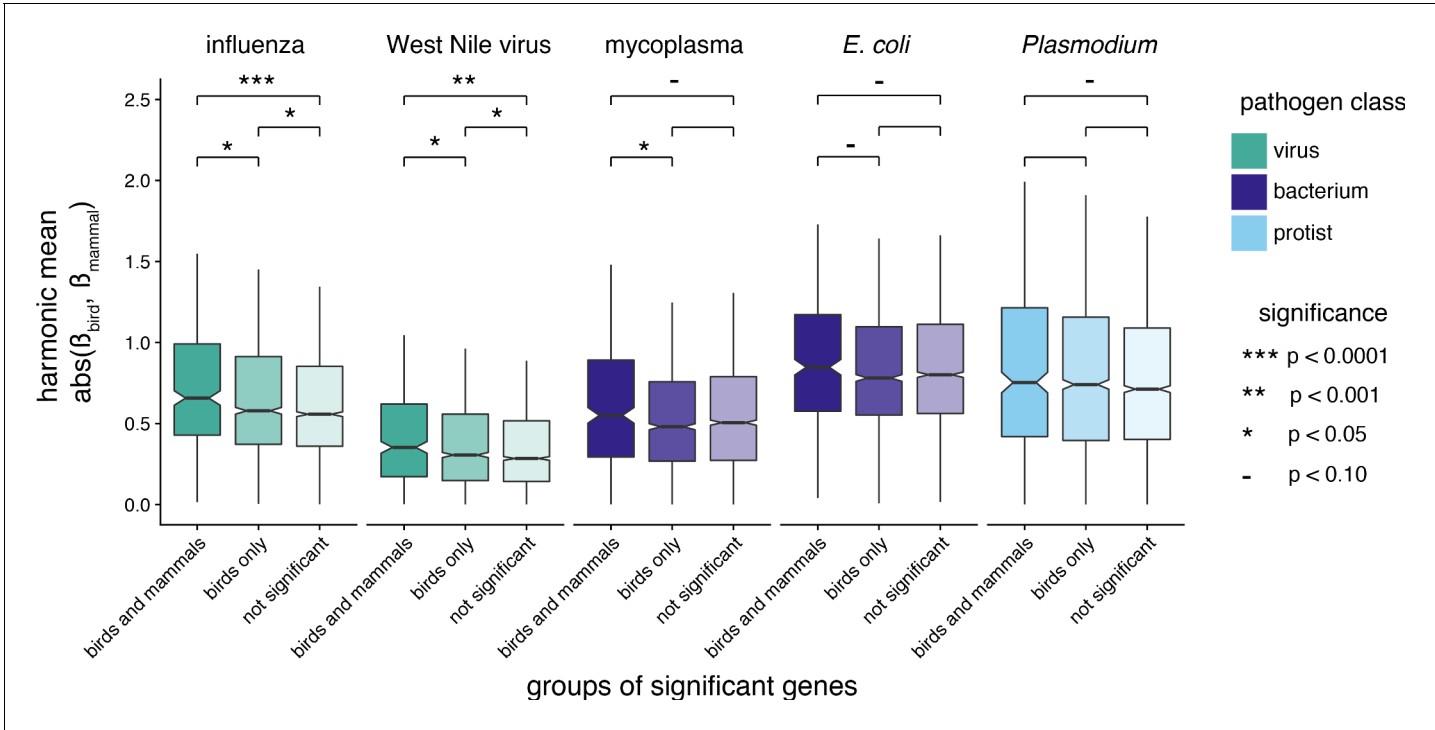

**Figure 8.** Comparison of differential expression effect values for groups of genes, across pathogens with transcriptome data available for both birds and mammals. Groups of genes are defined as being under positive selection in both birds and mammals, in birds only, or neither birds nor mammals (not significant). Differential expression effect values for each gene are calculated as the harmonic mean of the absolute β values of birds and mammals. We compared the mean of each category to that of the other two categories within each pathogen with Mann-Whitney U-tests, and the significance-level for each test is indicated by asterisks. Comparisons with p>0.10 are left blank. Note that boxplot outliers are not depicted.
DOI: https://doi.org/10.7554/eLife.41815.020

## Pathogens are a strong selective pressure in birds

Our pathway enrichment results and differential expression results imply that pathogens are one of the strongest selective pressures on amino acid sequence of protein coding genes in birds. Our site test results, which require the same sites to be under selection across species, suggest that host-pathogen interactions are constrained to target specific sites in the same genes in different species. Without these site constraints, genes may be under selection in many different lineages of birds, but there may also be much greater variation in molecular pathways under selection at more recent evolutionary timescales. Our PCAs of parameters of positive selection for each gene for each species support this hypothesis. Following PC1, axes of variation either separate clades of species (e.g. ratites, song birds), or specific lineages. Three previous studies that performed genome-wide scans for positive selection in specific bird lineages further support this hypothesis. First, *Nam et al., 2010* compared positive selection acting on three avian lineages, and while 1751 genes were evolving more rapidly than average in one of these three lineages, only 208 were common to all. *Backström et al., 2013* compared signatures of positive selection in two species of galliformes and two species of passerines, and found that only the passerine lineages showed GO enrichment with terms related to fat metabolism, neurodevelopment and ion binding. Finally, *Zhang et al., 2014b* found evidence for positive selection in the three vocal-learning bird lineages enriched for neural-related GO terms.

Previous studies of immune gene evolution in birds focus on receptor genes known to be hotspots of host-pathogen co-evolution, the Toll-like Receptors (TLRs) and Major Histocompatibility Complex (MHC). Five of the 10 avian TLRs are present in our dataset, with TLRs 1A, 1B, 2A, and 2B likely filtered out due to their recent duplication and TLR21 likely filtered out due to missing data caused by sequencing difficulty (*Alcaide and Edwards, 2011*; *Grueber et al., 2014*). For the five TLRs in our dataset we observe that the M0 ω values are similar to those observed by a previous

study (*Alcaide and Edwards, 2011*). In addition, our results confirm those of *Alcaide and Edwards, 2011*; *Grueber et al., 2014*, and *Velová et al. (2018)*, which showed TLR5 as having the highest proportion of selected sites, and the endosomal TLRs (TLR3, TLR7) as having the smallest proportion of selected sites. These similarities from independent studies focusing on just a few genes give us additional confidence in the results of our larger dataset. Unfortunately, the complexity of the MHC genes means that they were not included in our dataset. However, a recent survey of selection across birds found selection for both classes of MHC loci (*Minias et al., 2018*), indicating that our overall patterns of selection likely hold true.

Receptor genes clearly have signatures of pathogen-mediated selection but signaling pathways (e.g. ECM-receptor interaction and cytokine-cytokine receptor interaction) and downstream genes in immune pathway are also under selection in our dataset. Pathogens have evolved many ways to avoid the host immune response, sometimes at receptors, but sometimes at signaling molecules or genes involved in other cellular processes (*Finlay and McFadden, 2006*; *Randall and Goodbourn, 2008*; *Pichlmair et al., 2012*; *Quintana-Murci and Clark, 2013*; *Sironi et al., 2015*). The strong signatures of positive selection we observe in these alternative pathways and locations suggest that pathogens not only consistently target the same sites in receptor genes, but also the same sites within genes with other functions. The gene with the highest proportion of significant lineages in birds as estimated by aBS-REL in the Influenza A pathway is not for a receptor gene, but a signaling gene – the gene TRIF, also known as TICAM1. TRIF is recruited by TLR3, a viral sensing TLR in birds, and activates a set of molecules that culminates in the activate of IRF7 or NF-κB (*Santhakumar et al., 2017*). This gene is under selection in 61% of avian lineages, one of the highest proportions in our dataset, and highlights that genes beyond the classically studied MHC loci and TLRs may be interesting candidates for future studies on the ecology and evolution of host-pathogen co-evolution.

## Non-immune functional pathways under positive selection in birds

In addition to pathways directly related to immune function, pathways related to DNA replication and repair were also significantly enriched for positively selected genes (*Figure 3*). However, positive selection in these pathways may also be indirectly related to immune functions. First, viruses frequently subvert these pathways (e.g. mismatch repair, non-homologous end-joining) to promote their own replication cycle (*Chaurushiya and Weitzman, 2009*; *Luftig, 2014*). Long-term antagonistic selection between hosts and viruses to control these pathways could produce long-term signatures of positive selection, as in immune pathways. In addition, these pathways promote chromosomal stability and remove damaged DNA bases. Birds are known for their compact genomes that show greater than average chromosomal stability (*Zhang et al., 2014b*), and a surprising paucity of transposable elements (TEs) (*Cui et al., 2014*; *Zhang et al., 2014b*; *Kapusta and Suh, 2017*). One effect on genome structure during the insertion of transposable elements is genome rearrangement due to homologous recombination (*Kazazian, 2004*). *Cui et al. (2014)* hypothesized that homologous recombination may be responsible for purging transposable elements from the genome, and even observed a galliform hepadnavirus in the process of being removed via homologous recombination. Current host-pathogen evolutionary arms races between birds and TEs are also observed in woodpeckers and allies (Piciformes). There is evidence of different CR1 families expanding at least three different times within the order, and purifying selection for polymorphic TEs in three closely related woodpecker species (*Manthey et al., 2018*). Finally, the *Kapusta et al. (2017)* observation that the non-recombining W chromosome and regions near centromeres had the highest TE richness also suggest that homologous recombination may allow natural selection to more efficiently remove insertions that are deleterious in the population. Our pathway enrichment results support this hypothesis, and the similar dynamics of positive selection at specific sites as those observed with immune gene pathways suggest that birds may experience a form of host-pathogen co-evolution with TEs.

Two other groups of pathways were enriched for positively selected genes, although these pathways did not have significantly more genes under selection than expected based on median gene lengths, suggesting some caution in interpretation is warranted. The first category, lipid metabolism, includes the steroid hormone biosynthesis pathway and linoleic acid metabolism pathway. Steroid hormone biosynthesis is known to be related to diverse life history strategies in birds (*Hau et al., 2010*), and linoleic acid, more common in seed-rich diets, is related to thermoregulation and can

vary across habitats (*Ben-Hamo et al., 2013*; *Andersson et al., 2015*). These pathways could be under selection in the diverse set of species included in our dataset. However, these are also known to be factors modulating the immune system (*Koutsos and Klasing, 2014*), and pathogens are known to target cellular processes beyond the immune system (*Pichlmair et al., 2012*). Further study including additional life history characteristics may help distinguish between these two selective forces.

The second category, phototransduction, likely relates to different avian life histories and the visual needs associated with those life histories. The genetics of the avian visual system has traditionally focused on the evolution of the cone receptor genes, and specifically variation in the short wave sensitivity type one pigment, which has shifted multiple times between ultraviolet and short wavelengths throughout birds with a single amino acid change (*Odeen and Hastad, 2003*; *Ödeen and Håstad, 2013*). However, a comparison of retinal transcriptomes from owls, falcons, and hawks, groups that have visual systems adapted to low-light environments (e.g. nocturnal or crepuscular species) or with visual systems tuned for high visual acuity, found evidence for positive selection on phototransduction genes (*Wu et al., 2016*). The strong signal of positive selection across the broad array of species chosen for our dataset suggests that these genes may be broadly important across many species, and an in-depth analysis of species associated with specific visual needs may uncover additional information on the evolution of this important avian sensory system.

Finally, our PCA to identify groups of species that have the same genes under selection identified one principle component that separates species across the avian phylogeny (*Figure 4*), which is significantly correlated with body mass (*Figure 5*). By correlating the parameter estimates for each gene with body mass to identify the genes driving this correlation, we find that the FoxO signaling pathway and weakly, the cellular senescence pathway were associated using gene set enrichment (*Supplementary file 1* Table 8). Lifespan is one trait that is highly correlated with body mass (*Furness and Speakman, 2008*), FoxO proteins are linked to the aging process (*Martins et al., 2016*), and cellular senescence may be linked to lifespan through telomere dynamics (*Monaghan and Haussmann, 2006*; *de Magalhães and Passos, 2018*). Within a species, telomeres typically degrade as an organism ages, but few interspecific studies have found a correlation between telomere length and lifespan (*Monaghan and Haussmann, 2006*). However, a recent comparative study in birds showed that telomeres shortened more slowly in species with longer lifespans, and that these results are conserved within families (*Tricola et al., 2018*). A study of genes associated with telomeres in mammals did not find any correlation between the strength of positive selection at these loci and body mass (*Morgan et al., 2013*). Our results, and the unique pattern of telomere lengthening observed in birds may be an ideal system to study the evolution of telomere dynamics, and the molecular underpinnings of these processes. Finally, telomere length mediates lifespan and lifetime fitness, both of which are reduced due to chronic malaria infection (*Asghar et al., 2015*), and suggests that this lineage-specific signature of selection may also be related to pathogen-mediated selection.

## Conclusions and implications

Across birds, and more generally across tetrapods, there is a clear signal of positive selection acting on immune genes, whether against pathogens or transposable elements. Our results demonstrate that the same genes and potentially even the same codons may be shared targets of pathogens to subvert the immune response across not just species but also tetrapod Classes (mammals, birds). Genes with particularly strong evidence of selection may be good candidates for further study from a functional and ecological perspective, and could broaden perspectives on the ecology and evolution of immunity beyond MHC loci and TLRs typically examined. From an applied perspective, there is a great need to understand which proteins or genes in immune gene networks are important in pathogen resistance to improve breeding strategies in economically important species (e.g. poultry; (*Kaiser, 2010*). Our work is a first step in this direction, and we provide a rich resource for the examination of specific genes and pathways.

Here we have only considered positive selection at a broad scale. Combining these results with those from populations or specific clades within birds and mammals may provide new insights on similarities or differences in long and short-term selection. Pathogen load is the strongest driver of local adaptation in humans (*Fumagalli et al., 2011*) and viruses are important drivers of population adaptation in flies (*Early et al., 2017*). From a network perspective, functional gene pathways under

strong selection in humans are directly or indirectly involved in immunity (*Daub et al., 2013*). Given the signatures of host-pathogen co-evolution we observe across birds, we expect that pathogens are an important driver of recent adaptation in bird populations as well.

## Materials and methods

### Identification, alignment and filtering of avian orthologs

Avian orthologs were identified, aligned, and filtered by *Sackton et al. (2018)*. We provide a brief outline of the methodology here, but full details and computer code can be found in *Sackton et al. (2018)*. The program OMA v.1.0.0 (*Roth et al., 2008*; *Altenhoff et al., 2013*) was used to infer patterns of homology among protein-coding genes across 39 sequenced bird (*Figure 1*) and three non-avian reptile (*Alligator mississippiensis, Anolis carolinensis,* and *Chrysemys picta*) genomes. For each gene set, the longest transcript was selected to represent that protein in the homology search.

Once OMA had completed, alignments were built for each OMA-defined homologous group using MAFFT v.7.221 (*Katoh and Standley, 2013*), RRID:SCR_011811), and a HMM was built for each protein alignment using HMMER v. 3.1b hmmbuild (*Johnson et al., 2010*), RRID:SCR_005305). Each HMM was then used to search the full set of OMA input both to verify that the same proteins were recovered as belonging to a homologous group, and to assign unassigned proteins if possible. Finally, a graph-based algorithm was used to add gene models not assigned to any OMA group to the best match if possible. This produced a new set of homologous groups, which we use in the following analyses.

These 45,367 hierarchical orthologous groups, or HOGs, were filtered to retain 16,151 HOGs with sequences for at least four species. Protein sequences were aligned with MAFFT v. 7.245 (*Katoh and Standley, 2013*), and filtered in three steps. First, entire columns were excluded if missing in more than 30% of species, had sequence in fewer than 10 taxa, or was missing in two of the three of the main taxonomic groups (paleognaths, neognaths, or non-avian outgroups). Second, poorly aligned regions were masked according to *Jarvis et al. (2014)* using a sliding-window similarity approach. Third, columns were removed using the same criteria as the first round. Next, entire sequences were removed from each alignment if they were over 50% shorter than their pre-filtered length or contained excess gaps. Finally, entire HOGs were removed if they contained more than three sequences for any species, did not have more than 1.5x sequences for the given number of species present in the alignment, or were less than 100 base pairs long. Nucleotide sequences for all remaining HOGs were aligned with the codon model in Prank v. 150803 (*Löytynoja and Goldman, 2008*). In total, 11,247 HOGs remained after all alignment and filtering steps.

Guide trees for use in the tests of selection were constructed for each alignment with RAxML v. 8.1.4 (*Stamatakis, 2014*), RRID:SCR_006086) under a GTR + GAMMA substitution model, partitioned into codon positions 1 + 2 and 3, with 200 rapid bootstrap replicates and a maximum likelihood tree search. In cases where species had more than one sequence in the alignment, we included all copies to produce a gene tree for that HOG.

### Tests of selection

Once HOGs had been identified and filtered, we considered them as representatives for genes, and so will refer to them as genes. To identify positively selected genes, we compared models of nearly neutral evolution to those that included signatures of positive selection at a proportion of sites across lineages in the avian phylogeny. Sites under positive selection are defined as those with elevated nonsynonymous/synonymous substitution ratios ($\omega = d_N/d_S$) compared to the expectation under neutral evolution, $\omega = 1$. We used two different programs to identify genes with evidence for elevated $\omega$ values at specific sites across avian lineages. First, we used the site models (*Nielsen and Yang, 1998*; *Yang et al., 2000*) implemented in the program Phylogenetic Analysis by Maximum Likelihood v4.8 (PAML, RRID:SCR_014932; (*Yang, 2007*)) to calculate likelihood scores and parameter estimates for seven models of evolution (*Table 1*). Because some genes contained gene duplicates, we ran all analyses of selection on gene trees from all 11,247 genes, and separately on the species tree for 8,699 genes that had a maximum of one sequence per species. We used the species tree generated by OMA from *Sackton et al. (2018)* as the phylogenetic hypothesis. First, we fit the M0 model, which estimates a single $\omega$ for all sites in the alignment. We used the branch lengths

estimated with the M0 model as fixed branch lengths for subsequent models to decrease computational time. To identify genes with evidence of positive selection, we conducted likelihood ratio tests between neutral models and selection models (models with ω > 1). We compared likelihood scores from the M1a vs. M2a, M2a vs. M2a_fixed, M7 vs. M8, and M8 vs. M8a models (*Supplementary file 1* Tables 2,3) (*Nielsen and Yang, 1998*; *Yang et al., 2000*; *Wong et al., 2004b*). We computed p-values according to a $\chi^2$ distribution with two, one, two, and one degree of freedom respectively.

In addition to the site tests implemented in PAML, we used BUSTED (*Murrell et al., 2015*), a modeling framework implemented in the program HyPhy (*Pond et al., 2005*), RRID:SCR_016162), to identify genes with evidence of positive selection at a fraction of sites. BUSTED uses a model that allows branch-to-branch variation across the entire tree (*Murrell et al., 2015*). Similar to the PAML models, BUSTED uses a likelihood ratio test to compare a model including selection (ω >1 at a proportion of sites) with one that does not. We parsed all PAML and HyPhy results with computer code (*Shultz and Sackton, 2019*; copy archieved at https://github.com/elifesciences-publications/avian-immunity) and ran all downstream analyses in R. For both sets of tests, we used the Benjamini-Hochberg approach to correct for multiple testing (*Benjamini and Hochberg, 1995*) with the p.adjust function in the stats packages in R v.3.5 (*R Core Development Team, 2008*). We considered an FDR-corrected p-value less than 0.05 as evidence for positive selection in that gene for a given model comparison.

Finally, in addition to testing for selection at particular sites across bird lineages, we used the aBS-REL method in HyPhy with default parameters (*Kosakovsky Pond et al., 2011*) to detect which specific lineages showed evidence of selection for each gene. For each lineage, including both tip species and internal branches, aBS-REL estimates a p-value for the presence of positive selection. We considered both the raw p-value as well as a p-value corrected for multiple testing within each gene. Fewer lineages showed evidence of selection with an FDR-corrected p-value, but all subsequent results were qualitatively consistent with both sets of tests. For simplicity and because the stringent correction may remove biologically-interesting lineages with weak to moderate selection, we present the results using the number of lineages considered nominally significant without multiple-test correction. We also used a script to parse all aBS-REL results and run all downstream analyses in R (*Shultz and Sackton, 2019*).

Previous work has found that alignment errors can result in substantial false positives (*Markova-Raina and Petrov, 2011*). However, our strict alignment filtering strategy and use of the evolution-aware PRANK aligner minimizes the possibility that our results are solely false positives (*Markova-Raina and Petrov, 2011*). Recombination also can elevate ω estimates, but the M7 vs M8 model has been shown to be robust to recombination (*Anisimova et al., 2003*), and these results give us the highest proportions of positively selected genes we observe in our dataset (*Table 2*). Finally, despite observing high proportions of selected genes, the overall trend of gene-wide estimates of ω <<1 is consistent with patterns of purifying selection on coding regions of the genome (*Figure 2B*). Furthermore the similarity in estimated ω values between this study and previous studies in birds with different sets of genome sequences or the use of pairwise estimates between chicken and zebra finch (*Nam et al., 2010*; *Zhang et al., 2014a*) give us confidence that our results are robust.

## Gene annotation

We annotated genes for downstream enrichment analyses using chicken (*Gallus gallus* assembly version 4.0; G.K.-S. *Wong et al., 2004a*) and zebra finch (*Taeniopygia guttata* assembly version 3.2.4; *Warren et al., 2010*) NCBI gene IDs from sequences of those species included in the alignment of each gene. Of the 11,247 HOGs, 10,889 could be assigned to a chicken NCBI gene id, 10,364 could be assigned to a zebra finch NCBI gene id, 10,142 could be assigned to both, and 136 could not be assigned to a chicken or zebra finch NCBI gene ID. In order to test additional pathways available for mammalian species (see below), we converted chicken and zebra finch NCBI gene IDs to human (*Homo sapien*; GRCh38.p10) NCBI gene IDs using the R biomaRt package version 2.36.1 (*Durinck et al., 2005*; *Durinck et al., 2009*). For both avian species, we downloaded the ENSEMBL gene IDs, NCBI gene IDs, and human homolog ENSEMBL gene IDs for each gene using the ggallus_gene_ensembl (chicken genes, Gallus-gallus-5.0) and tguttata_gene_ensembl (zebra finch genes, TaeGut3.2.4) datasets. For humans, we downloaded the ENSEMBL gene IDs and NCBI gene IDs from the human hsapien_gene_ensembl (human genes, GRCh38.p10) dataset. We assigned each gene by first identifying all human ENSEMBL gene IDs and NCBI gene IDs that were chicken

orthologs, and filled in missing IDs with zebra finch annotations. In total, 9,461 out of 11,247 genes could be annotated with human NCBI gene IDs.

## Functional gene pathway enrichment for lineages under positive selection in birds

We looked for patterns of positive selection among groups of genes with similar functions using KEGG pathway enrichment tests (*Kanehisa and Goto, 2000*; *Kanehisa et al., 2012*), RRID:SCR_012773). We used our most conservative set of genes as our test set – those with FDR-corrected p-values less than 0.05 for all site tests (N = 1,521), including the m1a vs. m2a PAML model comparison, m2a vs. m2a_fixed PAML model comparison, m7 vs. m8a PAML model comparison, m8 vs. m8a PAML model comparison, and BUSTED analysis (see *Table 1* for PAML model descriptions). Because of the similarity between the model results using gene trees and species trees (see Results), we use the gene tree results as input to maximize the number of genes that could be included in a functional analysis. Preliminary analyses using the species tree results are qualitatively similar to those presented here.

To conduct KEGG pathway enrichment analyses, we used the 'enrichKEGG' command from clusterProfiler v. 3.8.1 (*Yu et al., 2012*), RRID:SCR_016884) from Bioconductor v. 3.7 (*Gentleman et al., 2004*), RRID:SCR_006442) with chicken as the reference organism. We used the genes included in both PAML and HyPhy analyses with NCBI gene IDs (N = 10,874) as the gene universe for enrichment tests. To ensure genes not present in the chicken genome, but present in other bird species were not biasing our results, we also performed the functional enrichment test using zebra finch as the reference organism. Finally, we performed a final enrichment test using human as the reference organism to test whether the expanded KEGG pathways of humans could provide insights beyond those available for chicken and zebra finch. We visualized the results using modified versions of the 'dotplot' and 'cnetplot' commands in clusterProfiler.

We also tested whether median gene length for each pathway could explain our observed enrichment results. We used alignment length as a proxy for gene length, and first conducted a logistic regression with the selection status for each gene (under selection in all tests or not) as the dependent variable and alignment length as the independent variable. We then used the resulting model to estimate the probability of a gene identified as being under selection for each pathway based on median alignment length. We multiplied that probability by the number of genes in a pathway to calculate the expected number of genes under selection based on length alone, and compared that number to our observed value using a Fisher's exact test.

## Clustering genes under selection among bird lineages

We used aBS-REL results to understand how groups of species that experience similar selective pressures might show evidence for positive selection for the same genes. To do this, we created matrices of the parameter estimates (proportion of sites under selection, ω) at each gene for each species. Because some ω estimates were at the boundary value, we set all ω estimates greater than 10,000 to 10,000. We also replaced any non-significant (p-value>0.05) parameter estimates with an ω value of 1, and proportion of sites under selection value of 0. We used these matrices to conduct principle components analyses (PCA) to cluster species by either the proportion of sites under selection or the log-transformed ω value of each species for each gene. We replaced any missing values with the mean parameter estimates for that gene, log-transformed ω values, and performed the PCA with the prcomp function in R.

Only the first principle component grouped unrelated species consistently among parameters and gene trees or species trees (see Results), so we tested whether PC1 might be related to body mass, a measurement correlated with many life history characteristics (*Pienaar et al., 2013*). We extracted body mass measurements from each species using the CRC Handbook of Avian Body Masses (*Dunning, 2009*) and used phylogenetic generalized least squares (PGLS) (*Martins and Hansen, 1997*) to test for a correlation between the PC1 scores and log-transformed body mass. To obtain branch lengths for our species tree topology, we randomly selected one gene with one sequence for all species and used the branch lengths as calculated by the M0 model in PAML. Our results were robust to tests with alternative genes. We ran the PGLS analysis in R with the gls function from the nlme package 3.1–137 (*Pinheiro et al., 2013*), RRID:SCR_015655), with a both a

Brownian motion (*Felsenstein, 1985*) and an Ornstein-Uhlenbeck (*Hansen and Martins, 1996*) model of evolution. A Brownian motion fit better than the Ornstein-Uhlenbeck model (AIC > 2), so we report those results. However, the results are qualitatively similar. We visualized the two traits and the phylogeny using the 'phylomorphospace' function and the evolution of PC1 on the phylogeny using the a modified version of the 'plotBranchbyTrait' function from phytools v. 0.6–44 (*Revell, 2012*), RRID:SCR_015502).

To better understand which genes and molecular functions were contributing to the correlation between PC1 and body mass (see Results), we calculated a p-value for the association between the proportion of sites under selection and log-transformed body mass, and log-transformed ω values and log-transformed body mass for each gene separately. Due to the non-normal distribution of parameter estimates, we used Spearman's rank correlation with the cor.test function from the stats package in R (*R Core Development Team, 2008*). Although Spearman's rank correlation does not include phylogenetic correction, the aBS-REL p-values are estimated independently for each branch, and so should not be biased by phylogeny. We used the Benjamini-Hochberg approach to correct for multiple testing (*Benjamini and Hochberg, 1995*).

We tested whether there might be any functional signal in these genes using gene set enrichment with the Spearman's rank correlation values (ρ) as the input for each gene. To avoid biases in genes with only one or a few lineages under selection, we only tested genes with at least five lineages under selection (preliminary results with alternative cutoff suggest that results are robust to the specific cutoff used). We tested for gene set enrichment with the chicken KEGG pathways using the 'gseKEGG' command from clusterProfiler (*Yu et al., 2012*).

## Comparisons of avian and mammalian selection datasets

In order to identify shared signatures of selection in both birds and mammals, we compared our results to those of *Enard et al., 2016*. We used our BUSTED results as calculated using the species tree to ensure our results were comparable to their BUSTED tests of positive selection. We combined our datasets using the human ENSEMBL gene ID annotations (conversion methods described above). In total, we could identify 4,931 orthologous genes with results from both datasets. With the set of genes included in both studies, we re-calculated FDR-corrected p-values, and compared the proportion of genes significant in both birds and mammals with a p-value cutoff of 0.1, 0.01, 0.001 and 0.0001 to understand whether genes under weak or strong selection might produce different signals. We calculated significance of an increased overlap in genes under selection in both birds and mammals with a Fisher's exact test. We repeated this analysis after removing the 20% most constrained genes, defined here as the 20% of genes with the lowest m0 model ω values, to ensure that results were not driven by the reduced power to detect selection in constrained genes. We also tested whether an overlap in genes significant in both clades could be explained by gene length. To do this, for each p-value cutoff, we conducted a logistic regression with the selection status in mammals (significantly under selection or not according to the p-value cutoff) as the response variable, and the selection status in birds, alignment length, and their interaction as independent variables.

We tested whether pathogen-mediated selection might be an important factor in driving the overlap of these genes using KEGG pathway enrichment. We ran these tests as described above, with the genes under positive selection in both birds and mammals as the test set of genes, and the set of genes under selection in birds as the background set of genes. We used the four different FDR-corrected p-value cutoffs (p<0.1, 0.01, 0.001, or 0.0001) to identify genes under selection in each clade. Finally, we used permutation tests to ensure that our pathway enrichment results were not biased toward genes commonly under selection in birds. We randomly created test sets the same size as those empirically defined from the set of genes significant in birds and performed KEGG pathway enrichment. We calculated the enrichment score (proportion of selected genes in the pathway compared to the proportion of selected genes in the dataset) for each pathway significant in our bird-only results (described in above section) and included in our test of empirical data. That is, the pathway had to be significant in birds, and contain at least one gene under selection in both birds and mammals. We performed each permutation for each p-value cutoff 1000 times to generate a null distribution of enrichment values to compare to our empirical results.

## Association of genes under positive selection to pathogen-mediated transcriptional responses in birds

We independently tested whether genes under positive selection throughout birds were associated with pathogen-mediated immune responses using publicly-available transcriptome data. We tested whether genes that were differentially expressed in response to a pathogen challenge were more likely to be under positive selection. We identified 12 studies of birds that compared the transcriptomes of control individuals and individuals experimentally infected with a virus, bacterium or protist (*Supplementary file 1* Table 11; (*Smith et al., 2015a*; *Smith et al., 2015b*; *Sun et al., 2015a*; *Sun et al., 2015b*; *Videvall et al., 2015*; *Sun et al., 2016*; *Beaudet et al., 2017*; *Deist et al., 2017a*; *Deist et al., 2017b*; *Newhouse et al., 2017*; *Zhang et al., 2018*). We downloaded all available SRA files for each bioproject and extracted the fastq files with fastq-dump from SRA-Tools v. 2.8.2.1 (*Leinonen et al., 2011*). We used kallisto v. 0.43.1 (*Bray et al., 2016*), RRID:SCR_016582) to quantify transcript abundance with 100 bootstrap replicates. We used paired-end or single-end mode (assuming average fragment lengths of 250 base pairs with a standard deviation of 50 base pairs) as appropriate for each bioproject, using the ENSEMBL transcriptome reference for each species (*Supplementary file 1* Table 11). One species did not have an ENSEMBL reference available (*Spinus spinus*), so we mapped to the transcriptome reference of the closest available reference, *Serinus canaria*, downloaded from NCBI.

We tested for differential expression between experimentally infected individuals and control individuals with sleuth v0.30 (*Pimentel et al., 2017*), RRID:SCR_016883). In cases where individuals were available at different timepoints, had different phenotypes (e.g. resistant or susceptible), used different pathogen strains, or sequenced transcriptomes from different organs, we tested each condition separately. We considered a gene to be significantly differentially expressed if it had a q-value less than 0.05 and an effect size, quantified as the absolute value of $\beta$, greater than 1. We then combined results for each condition of each bioproject. We considered a gene to be differentially expressed for that study if it was significant in half of conditions defined as different time points and phenotypes for each organ (*Supplementary file 1* Table 11). For three studies, only a single condition had any appreciable signal, we use used a relaxed our cutoff to count a gene as significant if it was significantly differentially expressed in any condition (*Supplementary file 1* Table 11).

To compare genes across species, we translated all ENSEMBL gene IDs to homologous chicken ENSEMBL gene IDs from the R biomaRt package version 2.36.1 (*Durinck et al., 2005*; *Durinck et al., 2009*), RRID:SCR_002987), except for *S. canaria*, which we translated to chicken gene IDs by mapping genes IDs to sequences in the same gene alignments in our dataset. Some pathogens were represented by more than one study in our dataset. To combine the results for each pathogen, we considered each gene to be significant for that pathogen if it was significant in at least one study. We used logistic regression to test whether genes that were up-regulated (compared to no difference in transcription) were more likely to be under selection in birds (defined in above section), and to test whether genes that were down-regulated were more likely to be under selection in birds.

## Comparisons of gene expression patterns and positively selected genes in birds and mammals

Finally, we compared the pathogen-mediated transcriptional responses in birds to those in mammals. We used 14 previously published studies that generated transcriptomes for control individuals and pathogen-challenged individuals to identify differentially expressed genes in response to pathogen infection for a species of mammal (*Supplementary file 1* Table 11; (*Qian et al., 2013*; *Langley et al., 2014*; *Ogorevc et al., 2015*; *Rojas-Peña et al., 2015*; *DeBerg et al., 2016*; *Lee et al., 2016*; *Tran et al., 2016*; *Chopra-Dewasthaly et al., 2017*; *de Jong et al., 2018*). We chose studies that used similar pathogens as those used in the avian experiments to compare the expression profiles of the two clades as closely as possible, while acknowledging that such matching will necessarily be somewhat imprecise. We used the same preprocessing steps as described in the above avian transcriptomic section. In two studies, seven and nine different timepoints were used, with a large number of individuals giving increased power to detect differentially expressed genes. For these two studies, we required genes to be significant in half of all timepoints as well as overall (all infected individuals compared to control individuals). We translated all non-human ENSEMBL gene IDs to human ENSEMBL gene IDs using biomaRt to compare results across all bird and

mammal species. Finally, for birds and for mammals, we summarized results for each gene for each infectious agent, considering a gene to be differentially expressed if it was differentially expressed in any study. Despite the smaller number of genes identified in the joint bird and mammal datasets, results comparing the enrichment in bird-only studies as described above were robust (results not shown), so we have confidence that our combined bird and mammal dataset captured the signal observed with birds alone.

With our combined bird and mammal dataset, we first tested whether genes up-regulated in infected birds were also likely to be up-regulated in infected mammals, or whether genes down-regulated in infected birds were also likely to be down-regulated in infected mammals. We used a Fisher's exact test to test whether the proportions of up- or down- regulated genes in both clades deviated from null expectations. Then, we combined the gene expression results with the significance results across birds and mammals. We sought to test whether genes that were under positive selection in birds were likely to be under positive selection in mammals and differentially expressed (either up- or down-regulated in both clades). To do this, for each pathogen, we used logistic regression with genes under selection in birds as the response variable (under selection or not), and the mammalian selection status (under selection or not), the differential expression status in birds (up- or down-regulated), and their interaction as predictor variables. Finally, due to the variety of experimental setups and small number of genes up- or down-regulated in both birds and mammals, we used a more sensitive test to test whether the absolute value of mammal and bird $\beta$ values were significantly higher in genes under selection in both lineages, or genes under selection in birds, compared to genes not detected as being under selection with our BUSTED site tests. A larger absolute value of $\beta$ implies larger magnitudes of differential expression, regardless of the direction of selection or q-value significance. To ensure the $\beta$ values were as comparable as possible among studies, we first standardized the $\beta$ values to have a mean of 0 and standard deviation of 1 for each study. Then, for pathogen replicates within birds and mammals, we used the maximum $\beta$ value observed as the bird or mammal $\beta$ value for that gene (results were robust if the mean $\beta$ value was used instead). For each gene, we calculated the harmonic mean of bird and mammal $\beta$ values, and used a Mann-Whitney U-test to test whether mean $\beta$ values were significantly different between genes under selection (q-value < 0.05) in birds and mammals and genes under selection in birds only, between genes under selection in birds and mammals and genes not under selection in either lineage, and between genes under selection in birds only and genes not under selection in either lineage.

## Acknowledgements

We thank Scott Edwards, Hopi Hoekstra, and John Wakeley for feedback on the project, as well as members of the Edwards Lab and Harvard Informatics Group. We thank Alison Cloutier with assistance with alignments and filtering, and Julia Yu for early discussion. The computations in this paper were run on the Odyssey cluster supported by the FAS Division of Science, Research Computing Group at Harvard University.

## Additional information

### Funding
The authors declare that there was no external funding received for this work.

### Author contributions
Allison J Shultz, Conceptualization, Data curation, Software, Formal analysis, Investigation, Visualization, Methodology, Writing—original draft, Project administration, Writing—review and editing; Timothy B Sackton, Conceptualization, Data curation, Software, Formal analysis, Methodology, Project administration, Writing—review and editing

### Author ORCIDs
Allison J Shultz  https://orcid.org/0000-0002-2089-4086
Timothy B Sackton  http://orcid.org/0000-0003-1673-9216

**Decision letter and Author response**
Decision letter https://doi.org/10.7554/eLife.41815.078
Author response https://doi.org/10.7554/eLife.41815.079

## Additional files

### Supplementary files

• Supplementary file 1. Supplemental tables, see README for descriptions of each table.
DOI: https://doi.org/10.7554/eLife.41815.021

• Transparent reporting form
DOI: https://doi.org/10.7554/eLife.41815.022

### Data availability

All data generated and analyzed during this study, including all source data for all figures are included in the manuscript, supporting files, Dryad repository (doi:10.5061/dryad.kt24554). All alignments used to perform analyses of PAML and HyPhy results are also available in the Dryad repository. Computing scripts for all analyses are available at https://github.com/ajshultz/avian-immunity; copy archieved at https://github.com/elifesciences-publications/avian-immunity.

The following dataset was generated:

| Author(s) | Year | Dataset title | Dataset URL | Database and Identifier |
|---|---|---|---|---|
| Shultz A, Sackton T | 2018 | Data from: Immune genes are hotspots of shared positive selection across birds and mammals | https://dx.doi.org/10.5061/dryad.kt24554 | Dryad, 10.5061/dryad.kt24554 |

The following previously published datasets were used:

| Author(s) | Year | Dataset title | Dataset URL | Database and Identifier |
|---|---|---|---|---|
| TB Sackton, P Grayson, A Cloutier, Z Hu, JS Liu, NE Wheeler, PP Gardner, JA Clarke, AJ Baker, M Clamp, SV Edwards | 2018 | Convergent regulatory evolution and the origin of flightlessness in palaeognathous birds | https://www.ncbi.nlm.nih.gov/bioproject/PRJNA433110 | NCBI Bioproject, PRJNA433110 |
| Sun H, Liu P, Nolan LK, Lamont SJ | 2015 | Avian pathogenic Escherichia coli (APEC) infection alters bone marrow transcriptome in chickens | https://www.ncbi.nlm.nih.gov/bioproject/?term=PRJNA279487 | NCBI Bioproject, PRJNA279487 |
| Sun H, Liu P, Nolan LK, Lamont SJ | 2015 | Thymus Transcriptome Reveals Novel Pathways in Response to Avian Pathogenic Escherichia coli (APEC) Infection | https://www.ncbi.nlm.nih.gov/bioproject/284293 | NCBI Bioproject, PRJNA288323 |
| Smith J, Burt DW, Bencina D | 2015 | Analysis of the host response of chickens infected with Infectious Bursal Disease Virus (IBDV) | https://www.ncbi.nlm.nih.gov/bioproject/?term=PRJEB7219 | NCBI Bioproject, PRJEB7219 |
| Smith J, Smith N, Le Yu, Paton IR, Gutowska MW, Forrest HL, Danner AF, Seiler JP, Digard P, Webster RG, DW Burt | 2015 | Infection of chicken with avian influenza H5N1 and H5N2 viruses | https://www.ncbi.nlm.nih.gov/bioproject/?term=PRJEB7213 | NCBI Bioproject, PRJEB7213 |
| Smith J, Smith N, Le Yu, Paton IR, Gutowska MW, Forrest HL, Danner AF, Seiler JP, Digard P, Webster RG, DW Burt | 2015 | Infection of duck with avian influenza H5N1 and H5N2 viruses | https://www.ncbi.nlm.nih.gov/bioproject/?term=PRJEB7215 | NCBI Bioproject, PRJEB7215 |
| Beaudet J, Tulman | 2017 | Transcriptional Profiling of the | https://www.ncbi.nlm. | NCBI Bioproject, |

| | | | | |
|---|---|---|---|---|
| ER, Pflaum K, Liao X, Kutish GF, Szczepanek SM, Silbart LK, Geary SJ | | Chicken Tracheal Response to Virulent Mycoplasma gallisepticum Strain Rlow | nih.gov/bioproject/?term=PRJNA394119 | PRJNA394119 |
| Deist MS, Gallardo RA, Bunn DA, Kelly TR, Dekkers JCM, Zhou H, Lamont SJ | 2017 | RNA-seq data from the trachea epithelial cells of two inbred chicken lines at three time points after challenge with lentogenic Newcastle disease virus | https://www.ncbi.nlm.nih.gov/bioproject/?term=PRJEB19318 | NCBI Bioproject, PRJEB19318 |
| Zhang J, Kaiser MG, Deist MS, Gallardo RA, Bunn DA, Kelly TR, Dekkers JCM, Zhou H, Lamont SJ | 2017 | RNA-seq for splenic gene expression in response to Newcastle disease virus challenge in two chicken lines with different disease resistance | https://www.ncbi.nlm.nih.gov/bioproject/?term=PRJEB21688 | NCBI Bioproject, PRJEB21688 |
| Deist MS, Gallardo RA, Bunn DA, Dekkers JCM, Zhou H, Lamont SJ | 2017 | Resistant and susceptible chicken lines show distinctive responses to Newcastle disease virus infection in the lung transcriptome | https://www.ncbi.nlm.nih.gov/bioproject/?term=PRJEB21760 | NCBI Bioproject, PRJEB21760 |
| Videvall E, Cornwallis CK, Palinauskas V, Valkiūnas G, Hellgren O | 2014 | Eurasian Siskin (Carduelis spinus) Blood Transcriptome | https://www.ncbi.nlm.nih.gov/bioproject/?term=PRJNA257687 | NCBI Bioproject, PRJNA257687 |
| Newhouse DJ, Hofmeister EK, Balakrishnan CN | 2016 | Taeniopygia guttata Raw sequence reads | https://www.ncbi.nlm.nih.gov/bioproject/?term=PRJNA352507 | NCBI Bioproject, PRJNA352507 |
| Langley RJ, Tipper JL, Bruse S, Baron RM, Tsalik EL, Huntley J, Rogers AJ, Jaramillo RJ, O'Donnell D, Mega WM, Keaton M, Kensicki E, Gazourian L, Fredenburgh LE, Massaro AF, Otero RM, Fowler Jr VG, Rivers EP, Woods CW, Kingsmore SF, Sopori ML, Perrella MA, Choi AMK, Harrod KS | 2014 | Integrative 'omic analysis of experimental bacteremia identifies a metabolic signature that distinguishes human sepsis from SIRS | https://www.ncbi.nlm.nih.gov/bioproject/?term=PRJNA254331 | NCBI Bioproject, PRJNA254331 |
| DeBerg HA, Zaidi MB, Khaenam P, Gersuk V, Linsley PS, Estrada-Garcia T | 2015 | Elucidating the etiology and molecular pathogenicity of infectious diarrhea by high throughput RNA sequencing | https://www.ncbi.nlm.nih.gov/bioproject/?term=PRJNA285798 | NCBI Bioproject, PRJNA285798 |
| Jong E, Hancock DG, Hibbert J, Wells C, Richmond P, Simmer K, Burgner D, Strunk T, Currie AJ | 2017 | RNA-sequencing of neonatal monocyte responses to E. coli and S. epidermidis | https://www.ncbi.nlm.nih.gov/bioproject/?term=PRJNA395845 | NCBI Bioproject, PRJNA395845 |
| Lee E-Y, Lee H-C, Kim H-K, Jang SY, Park S-J, Kim Y-H, Kim JH, Hwang J, Kim J-H, Kim TH, Arif A, Kim S-Y, Choi Y-K, Lee C, Lee C-H, Jung JU, Fox PL, Kim S, Lee J-S, Kim MH | 2015 | Transcriptome profiling of influenza virus-infected human bronchial epithelial cells | https://www.ncbi.nlm.nih.gov/bioproject/?term=PRJNA305099 | NCBI Bioproject, PRJNA305099 |
| Urbanowski M, Shaw M, Heinz S, Benner C, Chang M, Pache L, Al- | 2017 | Human monocyte-derived macrophage (MDM) cell transcriptome response to infection with H1N1, H3N2, and H5N1 | https://www.ncbi.nlm.nih.gov/bioproject/?term=PRJNA382632 | NCBI Bioproject, PRJNA382632 |

| brecht R, Garcia-Sastre A | | influenza virus. | | |
|---|---|---|---|---|
| Metreveli G, Heinz S, Benner C, Chang M, Pache L, Albrecht R, Garcia-Sastre A | 2017 | Mouse lung transcriptome response to infection with H1N1, H3N2, and H5N1 influenza virus. | https://www.ncbi.nlm.nih.gov/bioproject/?term=PRJNA385346 | NCBI Bioproject, PRJNA385346 |
| Chopra-Dewasthaly R, Korb M, Brunthaler R, Ertl R | 2016 | Transcriptional profiling of mammary gland and spleen from Mycoplasma agalactiae infected sheep | https://www.ncbi.nlm.nih.gov/bioproject/?term=PRJEB17676 | NCBI Bioproject, PRJEB17676 |
| Ogorevc J, Mihevc SP, Hedegaard J, Bencina D, Dove P | 2011 | Transcriptomic response of goat mammary epithelial cells to Mycoplasma agalactiae challenge - a preliminary study | https://www.ncbi.nlm.nih.gov/bioproject/?term=PRJNA143425 | NCBI Bioproject, PRJNA143425 |
| Swann J, Winzeler E, Lewis N, Abdel-Haleem AM, Li S | 2016 | Dual RNA-sequencing of host and pathogen during Plasmodium berghei infection of hepatocytes in vitro | https://www.ncbi.nlm.nih.gov/bioproject/?term=PRJNA314450 | NCBI Bioproject, PRJNA314450 |
| Wellcome Trust Sanger Institute | 2013 | Simultaneous_host_parasite_RNA_sequencing_of_rodent_malaria_infection_time_course_ | https://www.ncbi.nlm.nih.gov/bioproject/?term=PRJEB1284 | NCBI Bioproject, PRJEB1284 |
| Wellcome Trust Sanger Institute | 2014 | Simultaneous_host_parasite_RNA_sequencing_of_mosquito_transmistted_rodent_malaria_in_two_host_backgrounds | https://www.ncbi.nlm.nih.gov/bioproject/?term=PRJEB4731 | NCBI Bioproject, PRJEB4731 |
| Tran TM, Jones MB, Ongoiba A, Bijker EM, Schats R, Venepally P, Skinner J, Doumbo S, Quinten E, Visser LG, Whalen E, Presnell S | 2016 | Molecular Hallmarks of Naturally Acquired Immunity to Malaria | https://www.ncbi.nlm.nih.gov/bioproject/?term=PRJNA227074 | NCBI Bioproject, PRJNA227074 |
| Rojas-Peña ML, Vallejo A, Herrera S, Gibson G, Arévalo-Herrera M | 2015 | Transcription Profiling of Malaria-Naïve and Semi-Immune Colombian Volunteers in a Plasmodium vivax Sporozoite Challenge [RNA-Seq] | https://www.ncbi.nlm.nih.gov/bioproject/?term=PRJNA279199 | NCBI Bioproject, PRJNA279199 |
| Qian F, Chung L, Zheng W, Bruno V, Alexander R, Wang Z, Wang X, Kurscheid S, Zhao H, Fikrig E, Gerstein M, Snyder M | 2012 | Identification of genes critical for resistance to infection by West Nile virus using RNA-Seq analysis | https://www.ncbi.nlm.nih.gov/bioproject/?term=PRJNA174747 | NCBI Bioproject, PRJNA174747 |

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
