## [Decision Letter]

Thank you for submitting your article "Immune genes are hotspots of shared positive selection across birds and mammals" for consideration by *eLife*. Your article has been reviewed by three peer reviewers, one of whom is a member of our Board of Reviewing Editors, and the evaluation has been overseen by a Reviewing Editor and Diethard Tautz as the Senior Editor. The following individual involved in review of your submission has agreed to reveal his identity: David Enard (Reviewer #2).

The reviewers have discussed the reviews with one another and the Reviewing Editor has drafted this decision to help you prepare a revised submission.

Summary:

The reviewers find that your study on parallel adaptive evolution in birds and mammals is interesting and is a solid contribution to the field. In particular, they appreciated the scope of the study, the methods used, the extent of the taxonomic diversity considered and the insight into how pathogens may drive the evolution of genes and proteins in similar ways in different groups of vertebrates. One reviewer raised issues regarding the novelty of the work and noted that the novelty over previous observations could be better addressed in the Introduction.

The reviewers also raised a number of concerns that must be adequately addressed before the paper can be accepted. Some of the required revisions will likely require further analysis and interpretation but do not involve extensive addition to the analyses already performed.

Essential revisions:

1) While the dataset is impressive, and the analyses appear to be sound and well performed, one reviewer found it difficult to see what would justify the publication of this work in *eLife* because he perceived a lack of significant advancement over previous work. It would be important to know why such additional investigations have to be undertaken in the Introduction and state what was missing from previous investigations.

2) Please make sure that all the results of all selection tests conducted are made available. PAML, aBS-REL and BUSTED p-values, model parameter estimates, etc.

3) One of the main concerns was related to Figure 4A and the convergence of adaptation signals between mammals and birds. First, it is excellent that the authors used BUSTED for this analysis, because it is sensitive to both site and branch level signals of positive selection. The claim is therefore not restricted in its scope. This is related to the fact that BUSTED has reduced statistical power in highly constrained proteins where there are fewer sites that can adapt. Because highly constrained proteins are likely to be highly constrained in both mammals and birds, this means that the proteins where BUSTED will struggle to detect adaptation because of elevated constraint are the same in mammals and birds. These coldspots of lower statistical power could create an overall convergence of adaptation that reflects heterogeneity in levels of constraint/purifying selection rather than shared, active targeting of hosts and pathogens. That said, the odds ratios reported in Figure 4A are remarkably high and thus most likely due to actual hotspots of positive selection and not due to coldspots. It would still add to the strength of the results to firmly exclude that coldspots of increased purifying selection are a driving factor. To do this, the authors could reproduce the analysis in Figure 4A but this time excluding the most highly constrained orthologs. The authors could for example exclude the top 20% genes with the highest constraint, corresponding to the 20% genes with the lowest overall omega or alternatively omega-zero (constrained class).

4) Another main concern relates to the analysis in subsection “Lineages clustered by genes under selection in birds are most strongly related to body size and lifespan”, second paragraph. The magnitude of p-values does not necessarily indicate that the parameter estimates for omega are high, or that the estimate for the proportion of sites under positive selection is high. These variables seem more interesting than whether or not positive selection can be detected in a given gene. Related to this, it would be informative and interesting to assess variation among functional categories in gene length and whether this variation is correlated with the p-values for positive selection. Is it the case that it is more difficult to detect weak positive selection on a site in a small gene compared to a large gene? To the extent that gene length is correlated between orthologs in birds and mammals, this could also increase the overlap of positively selected genes within a functional category.

5) It would be useful to have a transparent and quantitative description of the effect size of the enrichments (e.g. "XXX of the positively selected immune related genes in birds would be expected to also be positively selected in mammals by chance, but in fact YYY of the immune related genes were positively selected in both clades.")

---

## [Author Response]

Essential revisions:1) While the dataset is impressive, and the analyses appear to be sound and well performed, one reviewer found it difficult to see what would justify the publication of this work in eLife because he perceived a lack of significant advancement over previous work. It would be important to know why such additional investigations have to be undertaken in the Introduction and state what was missing from previous investigations.

We have added several statements in the Introduction to more directly address gaps in the literature and how our study fills these gaps.

2) Please make sure that all the results of all selection tests conducted are made available. PAML, aBS-REL and BUSTED p-values, model parameter estimates, etc.

The results of all selection tests are available in our Dryad repository in the 01_input_raw_paml_hyphy_results. Collated versions of these results are available in the 01_output_processed_data folder, with all processing steps available in the 01_PAML_HyPhy_Res_DataPrep.R script. All analyses conducted in this paper, input and output files can be found in the subsequent input folders, output folders and scripts in the Dryad repository.

3) One of the main concerns was related to Figure 4A and the convergence of adaptation signals between mammals and birds. First, it is excellent that the authors used BUSTED for this analysis, because it is sensitive to both site and branch level signals of positive selection. The claim is therefore not restricted in its scope. This is related to the fact that BUSTED has reduced statistical power in highly constrained proteins where there are fewer sites that can adapt. Because highly constrained proteins are likely to be highly constrained in both mammals and birds, this means that the proteins where BUSTED will struggle to detect adaptation because of elevated constraint are the same in mammals and birds. These coldspots of lower statistical power could create an overall convergence of adaptation that reflects heterogeneity in levels of constraint/purifying selection rather than shared, active targeting of hosts and pathogens. That said, the odds ratios reported in Figure 4A are remarkably high and thus most likely due to actual hotspots of positive selection and not due to coldspots. It would still add to the strength of the results to firmly exclude that coldspots of increased purifying selection are a driving factor. To do this, the authors could reproduce the analysis in Figure 4A but this time excluding the most highly constrained orthologs. The authors could for example exclude the top 20% genes with the highest constraint, corresponding to the 20% genes with the lowest overall omega or alternatively omega-zero (constrained class).

We appreciate the reviewer comments here and have accordingly re-done the overlap analysis with the 20% of genes with the lowest omega excluded. The overlap results with these exclusions are extremely similar to the previous analysis, with the odds ratios being slightly higher for the larger q-value cutoffs of significance. We have included a version of Figure 4A with the most constrained genes excluded as Figure 4—figure supplement 1.

4) Another main concern relates to the analysis in subsection “Lineages clustered by genes under selection in birds are most strongly related to body size and lifespan”, second paragraph. The magnitude of p-values does not necessarily indicate that the parameter estimates for omega are high, or that the estimate for the proportion of sites under positive selection is high. These variables seem more interesting than whether or not positive selection can be detected in a given gene. Related to this, it would be informative and interesting to assess variation among functional categories in gene length and whether this variation is correlated with the p-values for positive selection. Is it the case that it is more difficult to detect weak positive selection on a site in a small gene compared to a large gene? To the extent that gene length is correlated between orthologs in birds and mammals, this could also increase the overlap of positively selected genes within a functional category.

We appreciate the perspective of the reviewers and had used p-values as the likelihood combined aspects of all parameters estimated. However, to make the analyses more intuitive for readers, we redid all analyses using the two parameters of interest in analyses of selection – the proportion of sites estimated to be under selection and omega values. We present both of these new analyses in the paper.

Regarding the second comment, the reviewers make an important point. Using logistic regression, we tested whether genes we identified as being under selection had different alignment lengths (as a proxy for gene length) on average compared to genes not under selection. We found that there was a significant relationship, and so incorporated this relationship into downstream analyses. Specifically, we added a new analysis of how average gene length varies in KEGG pathways. Using our logistic regression model, we calculated the probability that genes in that pathway would be under selection, based on the median gene length for a given pathway. We then used this probability to calculate the number of genes expected to be under selection in a given pathway. As a conservative test for whether or not we observed more genes than expected, we conducted a Fisher’s exact test between the expected number of genes under selection and our observed number. While all pathways we identify as enriched have more genes under selection than expected, some of the pathways we detect as being enriched for selection are not significantly enriched compared to these expectations. However, the immune system pathways that are the core of our results are significantly enriched beyond expectations from alignment length alone. We present these results and add a caveat in our Discussion section of the additional pathways.

We also explored whether alignment length could explain the overlap between genes under selection in both birds and mammals. We conducted a logistic regression with the selection status in mammals (significant or not) as the dependent variable, and the selection status in birds, alignment length and the interaction between the two as independent variables, and found that while alignment length is significant, the selection status in birds remains a significant predictor of selection status in mammals even after accounting for alignment length; the interaction is not significant. Thus, our overlap results appear to be robust to concerns about alignment length biases (for all FDR-corrected p-value cutoffs).

5) It would be useful to have a transparent and quantitative description of the effect size of the enrichments (e.g. "XXX of the positively selected immune related genes in birds would be expected to also be positively selected in mammals by chance, but in fact YYY of the immune related genes were positively selected in both clades.")

We have added the expected number of positively selected genes for the Influenza A pathway and Herpes simplex pathway to the Results section to facilitate the reader’s understanding of the enrichment test. We also added the expected number of significant genes to our odds-ratio comparisons of genes under selection in birds and mammals in Supplementary file 7.